# Spontaneous slow cortical potentials and brain oscillations independently influence conscious visual perception

Lua Koenig[1], Biyu J. He[1,2,3]*

**1** Neuroscience Institute, New York University Grossman School of Medicine, New York, New York, United States of America, **2** Departments of Neurology, Neuroscience & Physiology, Radiology, New York University Grossman School of Medicine, New York, New York, United States of America, **3** Department of Biomedical Engineering, New York University Tandon School of Engineering, New York, New York, United States of America

\* biyu.he@nyulangone.org

**Data Availability Statement:** All code used to run the analysis are available at https://doi.org/10.5281/zenodo.14236766. All the data relevant to plotting the figures are available at https://doi.org/10.5281/zenodo.14291607.

## Abstract

Perceptual awareness results from an intricate interaction between external sensory input and the brain's spontaneous activity. Pre-stimulus ongoing activity influencing conscious perception includes both brain oscillations in the alpha (7 to 14 Hz) and beta (14 to 30 Hz) frequency ranges and aperiodic activity in the slow cortical potential (SCP, <5 Hz) range. However, whether brain oscillations and SCPs independently influence conscious perception or do so through shared mechanisms remains unknown. Here, we addressed this question in 2 independent magnetoencephalography (MEG) data sets involving near-threshold visual perception tasks in humans using low-level (Gabor patches) and high-level (objects, faces, houses, animals) stimuli, respectively. We found that oscillatory power and large-scale SCP activity influence conscious perception through independent mechanisms that do not have shared variance. In addition, through mediation analysis, we show that pre-stimulus oscillatory power and SCP activity have different relations to pupil size—an index of arousal—in their influences on conscious perception. Together, these findings suggest that oscillatory power and SCPs independently contribute to perceptual awareness, with distinct relations to pupil-linked arousal.

## Introduction

A key characteristic of perception is its variability over time: we might notice a crack in a windshield one day, finding it distracting and hard to ignore, but on the next day, fail to notice the crack entirely, as it fades into the background as we focus on driving. This variability shows that conscious perception is not determined solely by the incoming sensory information; instead, it is molded by additional neural mechanisms intrinsic in the brain that vary over time [1,2].

A long-standing line of work using human fMRI has revealed that the constantly fluctuating, large-scale spontaneous activity of the brain plays a crucial role in shaping conscious

**Funding:** This work was supported by a U.S. National Institutes of Health grant (R01EY032085, to BJH) and an Irma T. Hirschl Career Scientist Award (to BJH). The funders played no role in the study design, data collection and analysis, decision to publish, or preparation of the manuscript.

**Competing interests:** The authors have declared that no competing interests exist.

**Abbreviations:** AUC, area under the curve; AUROC, area under the ROC; BW, bandwidth; CD, Coordinate Descent; CF, center frequency; d.v., decision variable; EEG, electroencephalography; ICA, independent component analysis; LC, locus coeruleus; LC-NE, locus coeruleus-norepinephrine; MEG, magnetoencephalography; ROC, receiver-operator curve; SCP, slow cortical potential; SVM, support vector machine.

perception in the visual [3], auditory [4], and somatosensory [5] modalities. However, the underlying dynamical mechanisms remain unclear. Using electroencephalography (EEG) and magnetoencephalography (MEG), a separate line of studies has discovered 2 distinct electrophysiological signatures of pre-stimulus spontaneous activity that influence conscious perception: power [6,7] and phase [8,9] of brain oscillations in the alpha (8 to 12 Hz) and beta (13 to 30 Hz) frequency ranges, on the one hand, and large-scale activity patterns of aperiodic activity in the slow cortical potential (SCP, <5 Hz) range, on the other hand [10–12]. However, currently, a key unanswered question is whether brain oscillations and SCPs act on perception independently or through a common set of mechanisms. Addressing this question would reveal whether there is a common final pathway for diverse spontaneous electrophysiological phenomena to exert their influences on perception or, alternatively, whether there may be a multitude of channels through which spontaneous neural dynamics shape perception.

In this study, we focus on the power of alpha and beta oscillations and investigate whether their influences on conscious visual perception share mechanisms with SCP or rather operate independently. Alpha oscillations have been extensively studied in the context of visual perception. In occipital regions, pre-stimulus alpha power predicts whether a near-threshold stimulus will be perceived or missed [6,13–17]. These findings fit well with the interpretation that alpha oscillations reflect fluctuating cortical excitability in sensory regions [18–20]. Importantly, these findings were also supported by recent work that separated the alpha oscillations from the 1/f power spectrum [21]. Alpha oscillations might also influence perception through attentional mechanisms, since fluctuations in alpha power can correlate with shifts in attention [6,22–24]. In addition, neuromodulatory systems influencing arousal, with consequences on sensory and perceptual processing [25,26], may operate through changes in alpha oscillations [27–30].

Although less extensively studied, beta oscillations have also been suggested to influence conscious perception, specifically via changes in visuospatial attentional engagement. For instance, modulation of selective attention was shown to be associated with long-range phase synchrony in the beta band within the frontoparietal attentional network, resulting in enhanced target processing and suppressed nontarget processing in a visual task [31]. Similarly, higher parietal beta power was found to predict the accuracy of perceptual choices [32]. Further, repetitive TMS over the frontal eye field in the beta range facilitated conscious visual perception in a near-threshold task [33]. Taken together, these studies show that alpha and beta oscillations may influence conscious perception through a variety of neural mechanisms.

In addition to oscillatory activity, aperiodic SCPs have also been shown to powerfully modulate the conscious perception and recognition of visual stimuli. In visual tasks, the phase of spontaneous SCP shifts prior to the onset of a threshold-level stimulus influences the likelihood of detection [34]. Recent studies have shown that the pre-stimulus large-scale activity pattern of the SCP predicts whether a low-level visual stimulus will be seen [10], whether a high-level visual stimulus will be recognized [12], and which perceptual content will reach awareness in a bistable perception task [35]. Additionally, SCPs can correlate with pupil-linked arousal [12], which influences sensory encoding and perceptual processing [30,36–38]. Importantly, both pre-stimulus SCP and pre-stimulus alpha power have been shown to influence the criterion in conscious perception tasks [6,12], further underscoring the question of whether they might act on perception through shared mechanisms.

Despite strong evidence suggesting the involvement of both types of spontaneous activity (brain oscillations and aperiodic SCP) in conscious perception, it is currently unknown whether they interact with each other or rather act independently to modulate visual perception. The current study aimed to address this question. Further, we aimed to elucidate these neurophysiological signals' relations to pupil size, a noninvasive proxy of moment-to-moment

arousal fluctuations [36,39]. We analyzed 2 previously collected MEG data sets [10,12], in which participants performed a threshold-level visual perception task involving either low-level (Gabor patches) or high-level (images containing objects, faces, houses, or animals) visual stimuli. We extracted perceptually relevant measures of pre-stimulus alpha power, beta power and SCPs, and examined whether their influences on perceptual outcome had shared variances. The results reveal that brain oscillations and SCPs influence conscious visual perception through independent mechanisms and that their perceptual influences have distinct relations to arousal-mediated mechanisms.

## Results

### Paradigm and behavior

To investigate the influence of pre-stimulus neural activity on conscious visual perception, we analyzed 2 previously collected MEG data sets [10,12]. In these tasks, subjects were presented with a visual stimulus, specifically, a low-level Gabor stimulus (which was either left- or right-tilting) in the "low-level data set" and a high-level stimulus (which was either an animal, a face, an object, or a house) in the "high-level data set." The stimuli were titrated to the individual subject's perceptual threshold. Subjects were instructed to indicate whether they had consciously detected the stimulus (low-level data set) or consciously recognized a meaningful content in the stimulus (high-level data set). Previous work using these 2 data sets has shown that pre-stimulus large-scale activity patterns in the SCP, in a 2-s window preceding stimulus onset, strongly influence whether a threshold-level low-level or high-level stimulus enters conscious awareness [10,12]. Here, we first show that pre-stimulus alpha and beta oscillatory powers in the same time window also influence conscious perception. We then investigate their relationships with pre-stimulus SCP activity's influence on perception (**Fig 1G**).

We first describe the behavioral patterns in both data sets. In the low-level data set, subjects were shown a brief-duration (33 to 66 ms), low-contrast Gabor patch and asked to indicate whether they had consciously seen the stimulus, and what its tilt was (**Fig 1A**). On each trial, the orientation of the Gabor patch was randomly chosen between 45° and 135°, and subjects were asked to provide a guess about its orientation even if they did not subjectively perceive it (i.e., a two-alternative forced choice question). A small number of catch trials were also included, in which no stimulus was presented. On average, subjects reported seeing the stimulus in an average of 48.9 ± 4.4% (mean ± SEM) of "real" (i.e., stimulus present) trials—suggesting that the stimuli were indeed presented at subjective threshold—and in 0.09 ± 0.29% of catch trials (**Fig 1B**). When they reported seeing the stimulus in real trials, they correctly identified the orientation in 96.8 ± 1.3% of trials. When they reported not seeing the stimulus, they correctly identified the orientation in 62.0 ± 2.7% of trials, which was still significantly above the chance level of 50% (one-sample $t$ test against chance level of 50%, $p < 0.001$), consistent with earlier studies [40,41]. The paradigm and behavioral patterns in this data set were described in detail in previous papers [10,42].

In the high-level data set, subjects were shown a brief (66 ms), low-contrast object stimulus and asked to report whether they subjectively recognized a meaningful content in the image (i.e., recognition experience) and what the category of the image was (i.e., categorization report; see **Fig 1C**). The object image was randomly chosen from one of 4 categories on each trial. The object image set consisted of 20 unique images, with 5 exemplars in each category. In addition, in a small percentage of trials, a scrambled image was shown, created by phase scrambling one exemplar image from each category (**Fig 1C**, inset, bottom row), and the scrambled image was presented at the same contrast as its original counterpart. The phase scrambling procedure destroyed any meaningful content in the original image but preserved

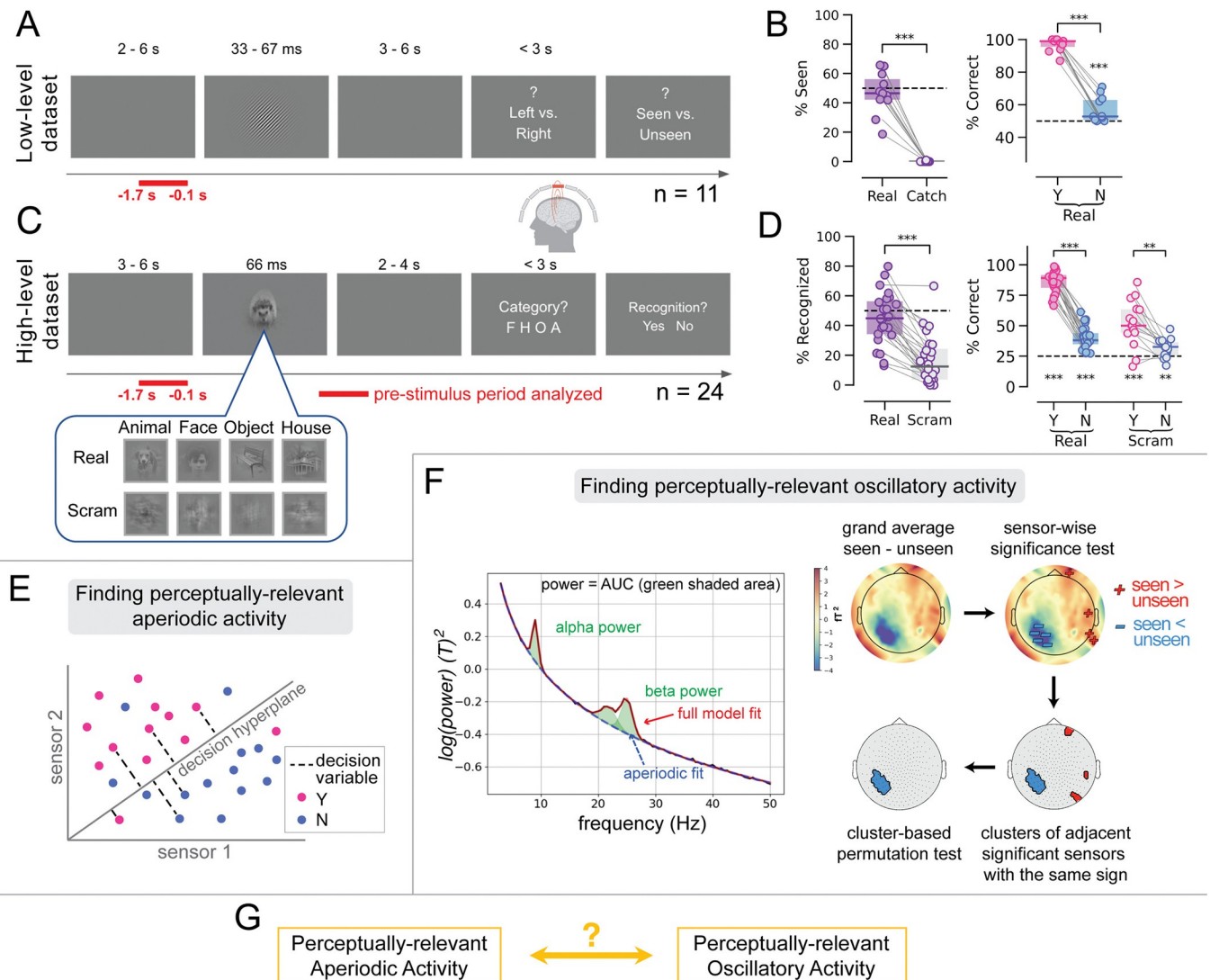

**Fig 1. Paradigm, behavior, and overall methods.** (**A**) Paradigm timeline for the low-level data set. Each trial began with a blank screen, after which a brief Gabor patch was presented for a duration determined by the participant's individual detection threshold. Subjects were asked to report the orientation of the stimulus and their detection experience. MEG activity in a 1.6 s window prior to stimulus onset (indicated by horizontal red bar) was used in further analyses. (**B**) Percentage of trials in which subjects reported detecting the stimulus, in real and catch trials. The dashed line represents the intended threshold-level detection rate (left panel). Percentage of trials in which subjects reported the correct orientation of the stimulus, for seen (Y, pink) and unseen (N, blue) trials. The dashed line represents the chance level of 50% for orientation discrimination (right panel). (**C**) Paradigm timeline for the high-level data set. Each trial began with a blank screen, after which a brief object stimulus was presented for 66 ms at a contrast determined by the participant's individual recognition threshold. Subjects were asked to report the category of the stimulus and their recognition experience. MEG was analyzed in a 1.6 s window prior to stimulus onset. (**D**) Percentage of trials in which subjects reported recognizing the stimulus for real and scrambled images. The dashed line indicates the intended threshold-level recognition rate for real images (left panel). Percentage of correctly categorized images, when stimuli were reported as recognized (pink) and unrecognized (blue), for real and scrambled images. The dashed line represents the chance level of 25% for categorization (right panel). In all box plots, the center line depicts the median and the edges indicate the quartiles. Stars above boxplots report paired tests, whereas stars below the dashed line report one-sample tests against a chance level of 25%; * $p < 0.05$, ** $p \leq 0.01$, *** $p \leq 0.001$. (**E**) Schematic representation of decoding from SCP activity. For each subject and pre-stimulus temporal window, the SCP activity from all seen or recognized (Y, pink) and unseen or unrecognized (N, blue) trials was projected onto a high-dimensional space (where each dimension corresponds to each MEG sensor, here represented for 2 hypothetical sensors). Multivariate decoders were trained to find the best hyperplane to separate trials based on perceptual outcomes. We used the distance of each trial from this decision hyperplane as the "SCP decision variable", or SCP d.v., plotted here with a dashed black line. (**F**) In the left panel, we obtained the power spectrum across the pre-stimulus temporal interval for each subject and extracted the oscillatory peaks in the alpha and beta ranges. We removed the aperiodic fit (dashed blue line) from the full power spectrum ("full model fit", red line) and computed the area under the curve (green shaded area) in alpha and beta bands to obtain their respective powers. This figure was adapted from the fooof toolbox [47]. In the right panel, we depict the topographic plots of the activity obtained by subtracting activity in unseen or unrecognized trials from the activity in seen or recognized trials. Wilcoxon signed-rank tests were then performed at each sensor. Adjacent sensors that showed significant differences with the same sign were clustered together. We determined which clusters were significant through cluster-based permutation. (**G**) The analyses will investigate whether perceptually relevant spontaneous aperiodic activity, specifically SCP d.v., correlates with perceptually relevant spontaneous

oscillatory activity, specifically alpha power and beta power. The data underlying this figure can be found as Fig 1_Data at https://doi.org/10.5281/zenodo.14291607. d.v., decision variable; MEG, magnetoencephalography; SCP, slow cortical potential.

low-level image features that have different statistical properties across categories. Therefore, reporting subjectively recognizing a meaningful content in a scrambled image constituted a false alarm. On average, subjects reported recognizing 44.9 ± 3.5% of real images (consistent with the thresholding procedure which aimed for 50% recognition rate) and 17.2 ± 3.3% of scrambled images (**Fig 1D**). The categorization accuracy in real image trials was 86.5 ± 1.8% for recognized trials and 40.1 ± 1.8% for unrecognized trials, which were significantly different (paired $t$ test, $p < 0.001$). For scrambled trials, we computed "categorization accuracy" according to the category of the original image used to create the scrambled image, which was 51.9 ± 4.7% in recognized trials and 31.6 ± 1.9% in unrecognized trials, with a significant difference between them ($p = 0.0015$). Categorization accuracy was significantly above chance level in all conditions (one-sample $t$ tests, all $p < 0.005$). In the case of scrambled-image trials, this means that low-level image features that differed between categories drove the participants' false alarm responses, suggesting that these false alarm responses reflected genuine false perceptions rather than mistaken button presses. The paradigm and behavioral patterns in this data set were described in detail in a previous paper [12].

In the following analyses, we investigate pre-stimulus brain activity predisposing the subject to have a conscious visual experience, as indicated by a "seen" response in the low-level data set and a "recognized" response in the high-level data set. Both because the previous literature has shown that the effects of interest (alpha/beta power and SCPs) predict criterion effects [6,10,12,43], which correspond to perceptual reports in both real and catch/scrambled trials, and to increase statistical power, the analyses use all trials in both data sets, including both real and catch/scrambled trials (catch trials accounted for approximately 5% of trials in the low-level data set; scrambled trials 16.7% of all trials in the high-level data set). As such, we focus on whether different pre-stimulus neural activities have shared influences on perceptual outcome, instead of dissociating pre-stimulus activity's influence on sensitivity and criterion related to conscious perception as previous studies have done [6,12].

## Whole-brain SCP activity patterns in the pre-stimulus period predict perceptual outcome

SCPs are the low-frequency (<5 Hz) component of the aperiodic fluctuations of brain field potentials that can be measured by EEG, MEG, or intracranial recordings [44]. To extract SCP activity, we first filtered MEG activity in the 0.05 to 5 Hz range [10]. Using this activity across the whole brain, we trained a classifier to discriminate between perceived and unperceived (i.e., seen versus unseen in the low-level data set, recognized versus unrecognized in the high-level data set) trials in the pre-stimulus interval spanning 1.7 to 0.1 s before stimulus onset. A simulation showed that at 0.1 s before stimulus onset, the data are unaffected by any temporal spearing of post-stimulus activity due to the frequency-domain filter applied (Fig A in **S1 Text**).

The decoder outputs a decision variable (d.v.), which is a continuously valued variable that measures the strength of evidence for each perceptual outcome, with positive values suggesting that the "seen/recognized" outcome is preferred and negative values suggesting that the "unseen/unrecognized" outcome is preferred (**Fig 1E**). Thus, for each trial, we obtained the SCP d.v., quantifying the strength of evidence for either perceptual outcome, at each time point within the pre-stimulus interval.

Consistent with the original publications showing successful pre-stimulus SCP decoding of perceptual outcome [10,12], the SCP d.v. timecourse averaged across trials and subjects was positive for perceived trials and negative for unperceived trials across the pre-stimulus intervals in both the low-level data set (**Fig 2A**) and the high-level data set (**Fig 2D**), suggesting that SCP contains significant predictive information about the outcome of conscious perception at the single-trial level.

## Pre-stimulus alpha power predicts perceptual outcome

To assess the impact of alpha power on conscious perceptual outcome, we first extracted the oscillatory power in the alpha frequency range (7 to 14 Hz), separate from the aperiodic activity (i.e., the oscillatory power above and beyond the 1/f power spectrum [45–47]), in four 400-ms time windows within the pre-stimulus time interval (−1.7 to −0.1 s). We then compared pre-stimulus alpha power in each time window between perceived and unperceived trials at every sensor (**Fig 1F**).

In the low-level data set, we obtained a significant sensor cluster distributed over posterior occipital cortex, in which the pre-stimulus alpha power was significantly lower in seen than unseen trials in the time interval from −1.7 to −1.3 s (cluster-based permutation test; $p < 0.001$, $W_{cluster} = -3,648$; **Fig 2B**). The effect in this sensor cluster persisted across all pre-stimulus time windows but became weaker over time.

In the high-level data set, we obtained 2 significant sensor clusters, in the time interval of −1.3 to −0.9 s ($p = 0.039$, $W_{cluster} = 8,092$) and −0.5 to −0.1 s ($p = 0.029$, $W_{cluster} = 9,256$), respectively (**Fig 2E**). Both clusters were distributed anteriorly and had higher pre-stimulus alpha power in recognized than unrecognized trials—which was in the opposite direction of the pattern in the low-level data set.

The alpha power cluster identified in the low-level data set reflects a well-documented effect in the literature, whereby pre-stimulus alpha power in occipital regions negatively predicts conscious detection of low-level threshold stimuli—an effect interpreted as reflecting alpha power's inverse relationship with cortical excitability in sensory regions [6,16,17,43]. Interestingly, the alpha power clusters in the high-level data set had an opposite effect and were located more anteriorly, suggesting that the relationship between alpha oscillatory power and conscious perception depends on the nature of the visual task. In the high-level data set, conscious recognition depends on the segmentation of a meaningful content, which may be less influenced by the excitability state of sensory regions [43]. Instead, we speculate that the effect seen in the high-level data set might reflect a positive relationship between alpha power and tonic alertness, which is regulated by the cingulo-opercular network located in more anterior regions [27,28], or the involvement of alpha oscillations in the top-down modulation of attention [48–50].

## Pre-stimulus beta power predicts perceptual outcome

We then conducted a similar analysis to assess the impact of beta (14 to 30 Hz) power on conscious perceptual outcome. In the low-level data set, we obtained 2 pre-stimulus beta power clusters, both in the time interval from −1.7 to −1.3 s (cluster-based permutation test; cluster 1: $p = 0.015$, $W_{cluster} = -1,678$; cluster 2: $p = 0.031$, $W_{cluster} = -1,330$; **Fig 3A**). Both clusters were distributed over posterior regions and had lower pre-stimulus beta power in seen trials than unseen trials.

In the high-level data set, we also obtained 2 beta power clusters, in the time interval of −0.9 to −0.5 s ($p = 0.027$, $W_{cluster} = 5,300$) and −0.5 to −0.1 s ($p < 0.001$, $W_{cluster} = 12,461$),

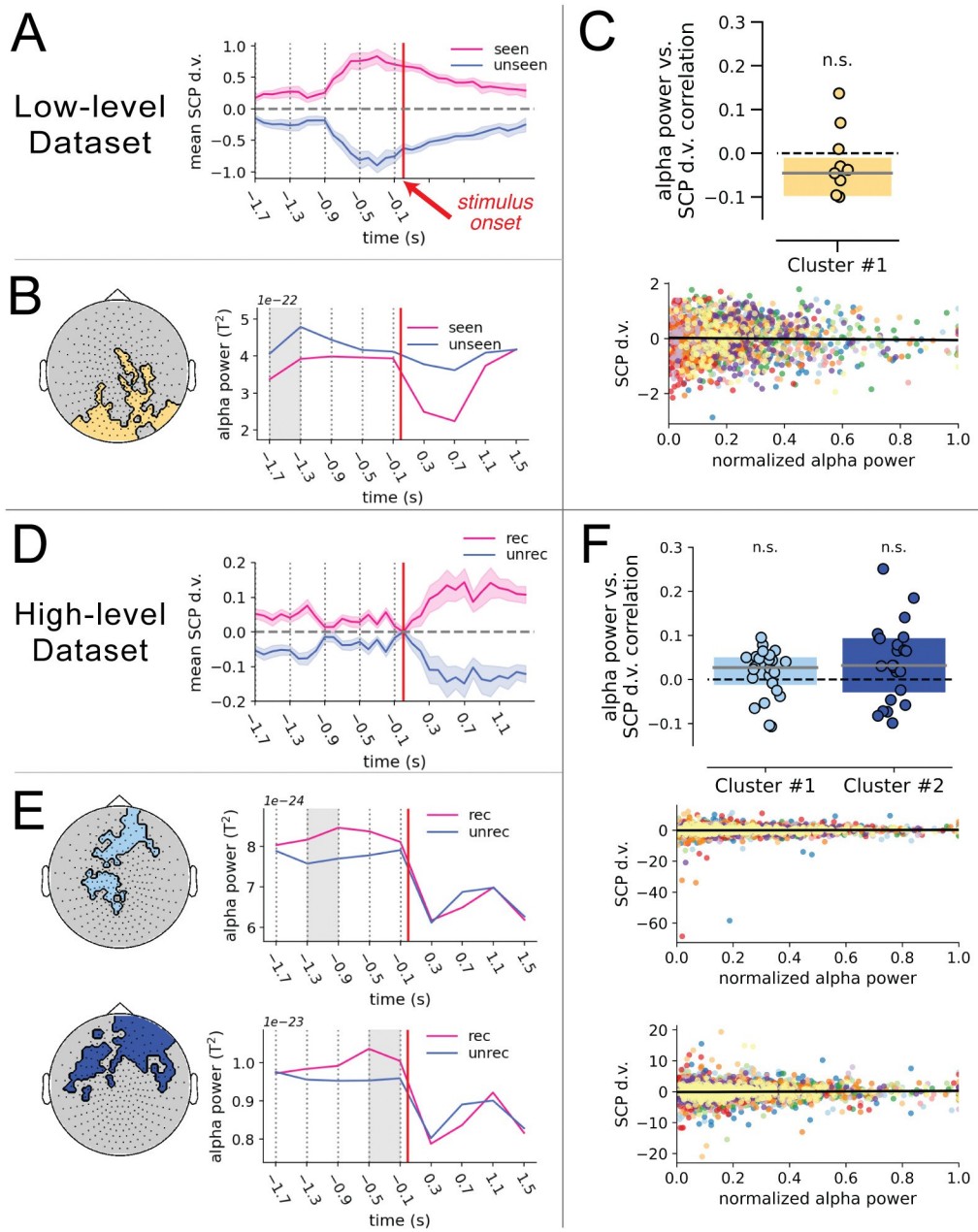

**Fig 2. Alpha power and SCP d.v. both influence perceptual outcome but do not covary.** (**A**) Timecourse of the mean SCP d.v. activity ($n = 11$) across the pre-stimulus interval for seen and unseen trials in the low-level data set. The dashed horizontal line represents an SCP d.v. of 0, indicating that the whole-brain SCP activity pattern does not distinguish between seen and unseen trials. The dotted vertical lines indicate the limits of the 4 temporal intervals considered during the analysis. The thick vertical red line indicates stimulus onset. Shaded areas indicate the paired sem. (**B**) Topographic plot of the alpha power cluster in the low-level data set (left). Timecourse of mean alpha power across the pre- and post-stimulus intervals for seen and unseen trials (right). The shaded gray area indicates the temporal interval in which this cluster was obtained. (**C**) Correlation between alpha power and SCP d.v. across single trials in the low-level data set. The top panel represents the boxplot of z-transformed correlation coefficients across subjects, tested against 0 with a Wilcoxon signed-rank test; n.s.: not significant. Dashed horizontal lines represent a null correlation. The bottom panel represents the scatterplot of normalized alpha power against SCP d.v. in single trials. Each color represents an individual subject ($n = 11$) and the bold black line represents the linear trend across subjects. (**D**) Timecourse of the mean SCP d.v. activity ($n = 24$) across the pre-stimulus interval for recognized (rec) and unrecognized (unrec) trials in the high-level data set. Shaded areas indicate the paired sem. (**E**) Topographic plots and timecourses for the 2 alpha power clusters obtained in the high-level data set. (**F**) Correlation between alpha power and SCP d.v. across single trials in the high-level data set. The top panel depicts the boxplots for z-transformed correlation

coefficients across subjects for each of the 2 alpha power clusters. The bottom panel depicts the scatterplots for the correlation for the first alpha power cluster (light blue cluster) and the second alpha power cluster (dark blue cluster). Each color represents an individual subject ($n$ = 24), and the bold black line represents the linear trend across subjects. The data underlying this figure can be found as Fig 2_Data at https://doi.org/10.5281/zenodo.14291607. d.v., decision variable; SCP, slow cortical potential.

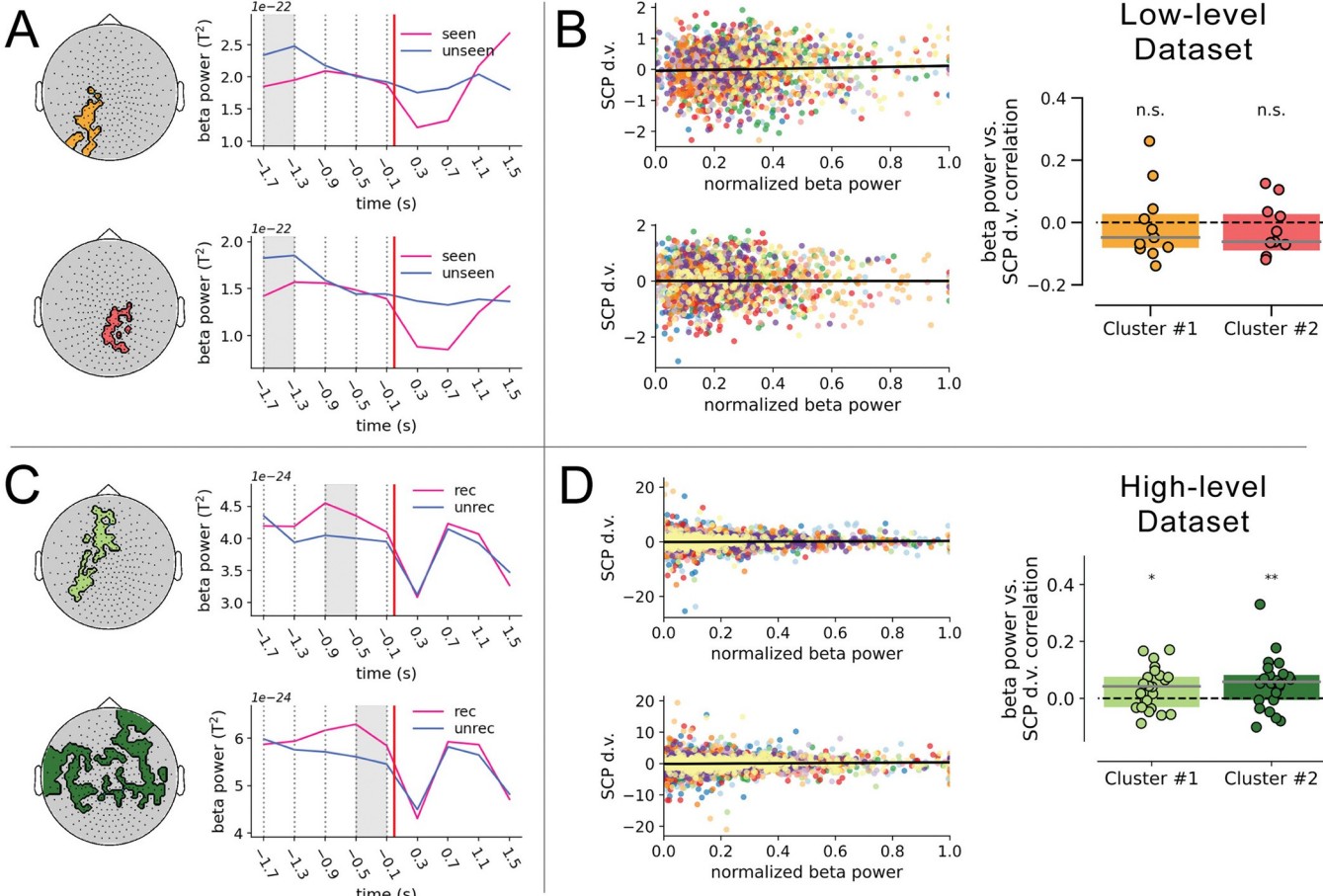

**Fig 3. Beta power and SCP d.v. both influence perceptual outcome and do not covary in the low-level data set but do in the high-level data set.** (**A**) The topographic plots of the first (orange) and second (red) beta power clusters in the low-level data set are depicted on the left, along with the corresponding timecourses of beta power across the pre- and post-stimulus intervals on the right, for seen and unseen trials. The dotted vertical lines indicate the limits of the 4 temporal intervals considered during the analysis. The thick vertical red line indicates the time of stimulus onset. The shaded gray area indicates the pre-stimulus temporal interval during which beta power significantly differed between seen and unseen trials for the corresponding cluster. (**B**) Correlation between beta power and SCP d.v. across single trials in the low-level data set. The left panel depicts the scatterplots corresponding to each cluster. Each color represents an individual subject ($n$ = 11), and the bold black line represents the linear trend across subjects. Boxplots on the right depict the z-transformed correlation coefficients across subjects for both clusters in the low-level data set, tested against 0 with a Wilcoxon signed-rank test; n.s.: not significant. Dashed horizontal lines represent a null correlation. (**C**) The topographic plots of the first (light green) and second (dark green) beta power clusters in the high-level data set are depicted on the left, along with the corresponding timecourses of beta power across the pre- and post-stimulus intervals on the right, for recognized and unrecognized trials. (**D**) Correlation between beta power and SCP d.v. across single trials in the high-level data set. The left panel depicts the scatterplots corresponding to each cluster. Each color represents an individual subject ($n$ = 24), and the bold black line represents the linear trend across subjects. Boxplots depict the z-transformed correlation coefficients across subjects for each of the 2 beta power clusters in the high-level data set, tested against 0 with a Wilcoxon signed-rank test. Dashed horizontal lines represent a null correlation. For all boxplots, the solid line represents the median, and the edges indicate the quartiles; * $p < 0.05$, ** $p \leq 0.01$. The data underlying this figure can be found as Fig 3_Data at https://doi.org/10.5281/zenodo.14291607. d.v., decision variable; SCP, slow cortical potential.

respectively (**Fig 3C**). Like for alpha power, both of these clusters were distributed more anteriorly and had higher pre-stimulus beta power in recognized than unrecognized trials.

Interestingly, the beta power clusters obtained in both data sets match the alpha power clusters in striking ways: in the low-level data set, both alpha and beta clusters had a more posterior location and had lower power preceding perceived trials; by contrast, in the high-level data set, both alpha and beta clusters had more anterior locations and had higher power preceding perceived trials. This suggests that the type of visual task modulates the specific impact of spontaneous brain oscillations in conscious perception.

## Alpha power and SCP independently influence conscious visual perception

In the previous analyses, we first obtained a single-trial metric of perceptually relevant spontaneous aperiodic activity, in the form of SCP d.v., which measures the strength of evidence in the SCP range for or against conscious perception (with positive values predicting conscious perception). We then obtained single-trial metrics of perceptually relevant spontaneous oscillatory activity, in the form of power fluctuations in sensor clusters where pre-stimulus alpha or beta power predicts conscious perceptual outcome. Our study's primary goal was then to assess whether aperiodic and oscillatory spontaneous activity have shared or independent influences on perception (**Fig 1G**).

To that end, we first assessed the relationship between perceptually relevant pre-stimulus SCP and alpha power activity. For each alpha power cluster (**Fig 2B** and **2E**), we averaged pre-stimulus alpha power in the relevant time window across sensors within the cluster and correlated this cluster-wide power metric with the SCP d.v. across trials, in the same time window. We found that in the low-level data set, the correlation between alpha power and SCP d.v. was not significant (Wilcoxon signed-rank test across subjects; $p = 0.15$, $W = -16$; **Fig 2C**). A Bayesian one-sample $t$ test yielded a Bayes Factor ($BF_{10}$) of 0.828. This can be interpreted as weak evidence in favor of the null hypothesis.

Similarly, in the high-level data set, the correlation between alpha power and SCP d.v. was not significant in either cluster ($p = 0.152$, $W = 99$; $p = 0.06$, $W = 83$; **Fig 2F**). A Bayesian one-sample $t$ test yielded a $BF_{10}$ of 0.331 for cluster 1 (light blue in **Fig 2E and 2F**), which is considered weak-moderate evidence in favor of the null hypothesis, and a $BF_{10}$ of 1.240 for cluster 2 (dark blue in **Fig 2E and 2F**), which is considered weak evidence in favor of the alternative hypothesis.

Together, these results suggest that although each spontaneous signal was individually linked to subsequent perceptual outcome, there was little evidence of a strong association between the signals, suggesting that their influences on perception are likely independent from each other.

## Beta power and SCP independently influence conscious visual perception

Next, we assessed whether pre-stimulus beta power and SCP have shared influences on conscious perception. To this end, similar to the previous analysis, we first correlated SCP d.v. and beta power (averaged across sensors within a cluster) across trials for each subject, and assessed the significance of the correlation at the population level.

In the low-level data set, the correlation between beta power and SCP d.v. was not significant in either cluster (Wilcoxon signed-rank tests; $p = 0.52$, $W = -25$; $p = 0.32$, $W = -21$; **Fig 3B**). A Bayesian one-sample $t$ test yielded a $BF_{10}$ of 0.3 for both clusters, representing moderate evidence in favor of the null hypothesis. This result suggests that beta power and SCP, although with significant individual influences on perceptual outcome, are not significantly associated with each other, and hence likely operate independently on perception.

In the high-level data set, we found a significant positive correlation between beta power and SCP d.v. in both clusters ($p = 0.037$, $W = 77$; $p = 0.009$, $W = 60$; Fig 3D). A Bayesian one-sample $t$ test yielded a $BF_{10}$ of 1.835 for cluster 1 and a $BF_{10}$ of 2.777 for cluster 2, which are considered weak evidence in favor of the alternative hypothesis. Given that beta power and SCP d.v. covaried in the high-level data set, we then investigated whether they had shared influences on perception. In principle, while shared perceptual influences require the covariation between 2 variables, it is important to note that such covariation does not necessarily indicate shared influences on perception.

To address this question, we constructed a linear regression model between beta power and SCP d.v., which allowed us to compute the residual of each signal when its shared variance with the other signal had been removed (Fig 4A, top). We then computed the ability of each

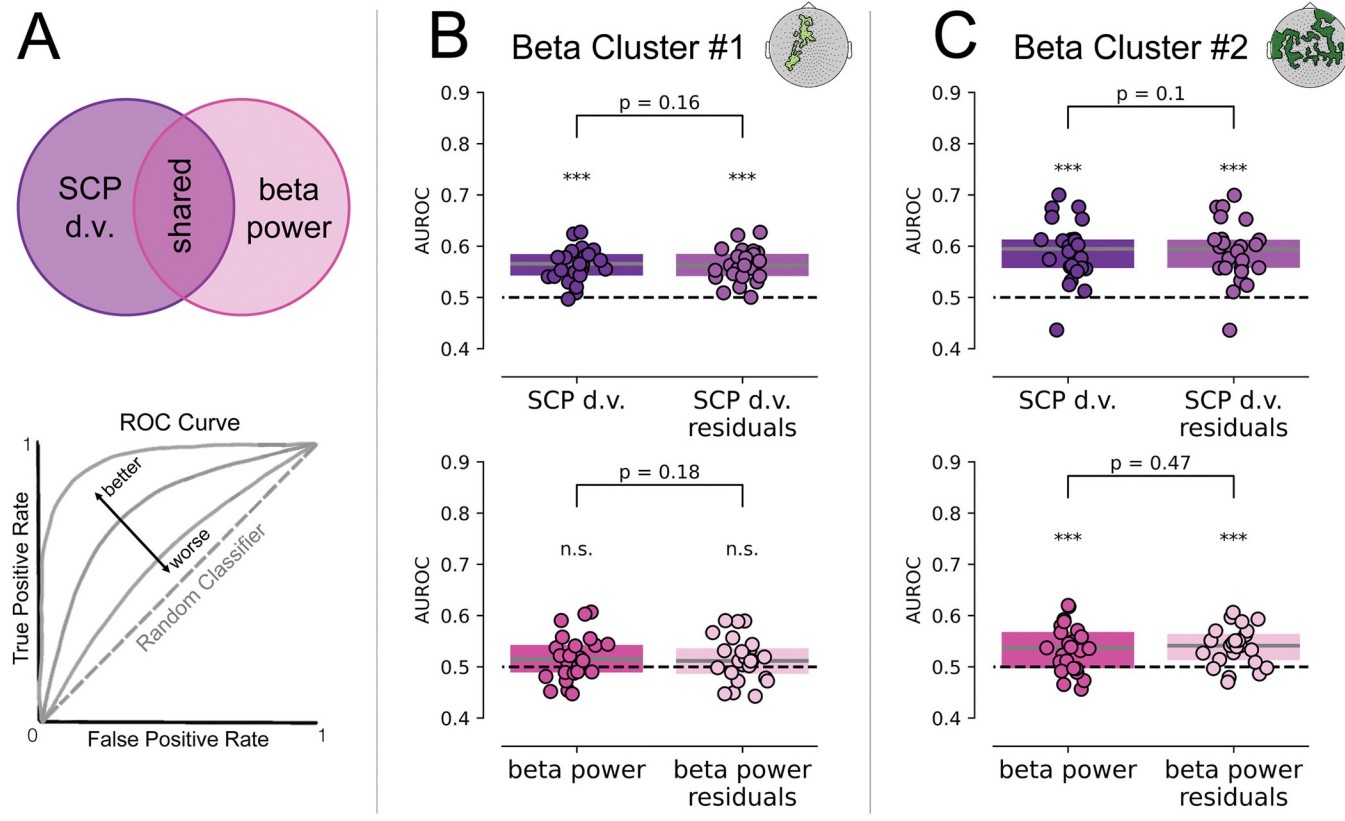

**Fig 4. AUROCs for beta power and SCP d.v. demonstrate that their influences on perceptual outcome are independent.** (**A**) Given that SCP d.v. and beta power were significantly positively correlated in the high-level data set, we modeled SCP d.v. and beta power as having shared variance, depicted by the overlap of the 2 circles in the top diagram. SCP d.v., including both independent variance and variance shared with beta power, is depicted in dark purple. SCP d.v. residuals, representing the variance in SCP d.v. once the shared variance with beta power has been removed, is depicted in light purple. Beta power, including both independent variance and variance shared with SCP d.v., is depicted in dark magenta. Beta power residuals, representing the variance in beta power once the shared variance with SCP d.v. has been removed, is depicted in light pink. The bottom panel represents a model ROC curve, in which the false positive rate (i.e., the rate at which a classifier will falsely label a trial as "recognized," based on a given activity metric) is on the X-axis and the true positive rate (i.e., the rate at which a classifier will correctly label a trial as "unrecognized," based on a given activity metric) is on the Y-axis. A random classifier will have an AUROC of 0.5, as depicted by the dashed gray line. Better classifiers will have AUROCs closer to 1. (**B**) AUROC scores for the first beta power cluster, comparing SCP d.v. (dark purple) and SCP d.v. residuals (light purple) when shared variance with beta power was removed (top panel) and comparing beta power (dark magenta) and beta power residuals (light pink) when shared variance with SCP d.v. was removed (bottom panel). (**C**) AUROC scores for the second beta power cluster, comparing SCP d.v. and SCP d.v. residuals (top panel) and beta power and beta power residuals (bottom panel). In B and C, dashed horizontal line represents the AUROC score of a random classifier. Scores were each compared to the random classifier value and to each other with Wilcoxon signed-rank tests. * $p < 0.05$, ** $p \leq 0.01$, *** $p \leq 0.001$, n.s.: not significant. The data underlying this figure can be found as Fig 4_Data at https://doi.org/10.5281/zenodo. 14291607. AUROC, area under the ROC; d.v., decision variable; ROC, receiver-operator curve; SCP, slow cortical potential.

signal, as well as its residual, to predict perceptual outcome using a receiver-operator curve (ROC) analysis (**Fig 4A**, bottom). The area under the ROC (AUROC) score from the ROC analysis quantifies the predictive power each signal has on perceptual outcome (see Methods). We then compared the AUROC scores between beta power and beta power residuals (once the shared variance with SCP d.v. had been removed) and between SCP d.v. and SCP d.v. residuals (once the shared variance with beta power had been removed).

This analysis showed that, in the first beta cluster (**Fig 4B**), SCP d.v. and SCP d.v. residuals AUROC scores were not significantly different (Wilcoxon signed-rank test; $p = 0.16$, $W = 92$, $BF_{10} = 0.875$), and both SCP d.v. and SCP d.v. residual had AUROC scores significantly above the chance level of 0.5 ($p < 0.001$, $W = 299$, $BF_{10} = 1.4e7$; $p < 0.001$, $W = 300$, $BF_{10} = 1.55e7$, respectively). Similarly, the AUROCs for beta power and beta power residuals were not significantly different ($p = 0.18$, $W = 102$, $BF_{10} = 0.536$). The AUROCs for beta power and beta power residual had trend-level significance when compared to 0.5 ($p = 0.0501$, $W = 208$, $BF_{10} = 2.021$; $p = 0.0604$, $W = 205$, $BF_{10} = 1.35$, respectively), which was likely due to the process of averaging beta power across sensors within the cluster, diluting the predictive power of beta power for perceptual outcome that was initially measured at the individual sensor level. Critical to our question, the AUROC scores did not differ between each signal and its residual, suggesting that removing the shared variance between beta power and SCP d.v. did not degrade either signal's predictive power for perception.

In the second beta cluster (**Fig 4C**), the AUROC scores for SCP d.v. and SCP d.v. residuals were not significantly different ($p = 0.1$, $W = 92$, $BF_{10} = 0.665$), and both were significantly higher than the chance level ($p < 0.001$, $W = 291$, $BF_{10} = 2e5$; $p < 0.001$, $W = 291$, $BF_{10} = 1.6e5$, respectively). Similarly, the AUROCs for beta power and beta power residuals were not significantly different ($p = 0.47$, $W = 124$, $BF_{10} = 0.308$), and both scores were significantly above the chance level ($p < 0.001$, $W = 257$, $BF_{10} = 66.6$; $p < 0.001$, $W = 278$, $BF_{10} = 1423$, respectively).

Together, these results show that, for both beta power and SCP d.v., when the variance they share with each other is removed, their ability to predict perceptual outcome is unchanged. This demonstrates that pre-stimulus beta power and SCP d.v. explain non-overlapping variance in the perceptual outcome, suggesting that they independently contribute to conscious perception.

## Alpha power and beta power have shared influences on perception

We next investigated whether alpha and beta power have shared influences on perception. To this end, we first investigated across-trial correlation between alpha power and beta power, using all pairs of significant clusters identified in the earlier analyses (**Fig 5**, top panels). Thus, we tested whether the pre-stimulus alpha power that predicted perception and the pre-stimulus beta power that predicted perception co-fluctuated from trial to trial. We found that alpha and beta power were positively correlated in all cluster pairs of both data sets (Wilcoxon signed-rank test; all $p < 0.001$; **Fig 5**, bottom panels).

To address whether alpha and beta power had shared influences on perceptual outcome, similar to the previous analysis between beta power and SCP, we computed the alpha power AUROC scores and compared them with the AUROC scores for alpha power residuals once its shared variance with beta power had been removed. Similarly, we compared AUROC scores for beta power and beta power residuals.

In the low-level data set (**Fig 5A**), we found that the scores for alpha power and alpha power residuals after removing variance from both beta clusters did not significantly differ (Wilcoxon signed-rank test; $p = 0.28$, $W = 1.08$). The AUROC scores were significantly higher

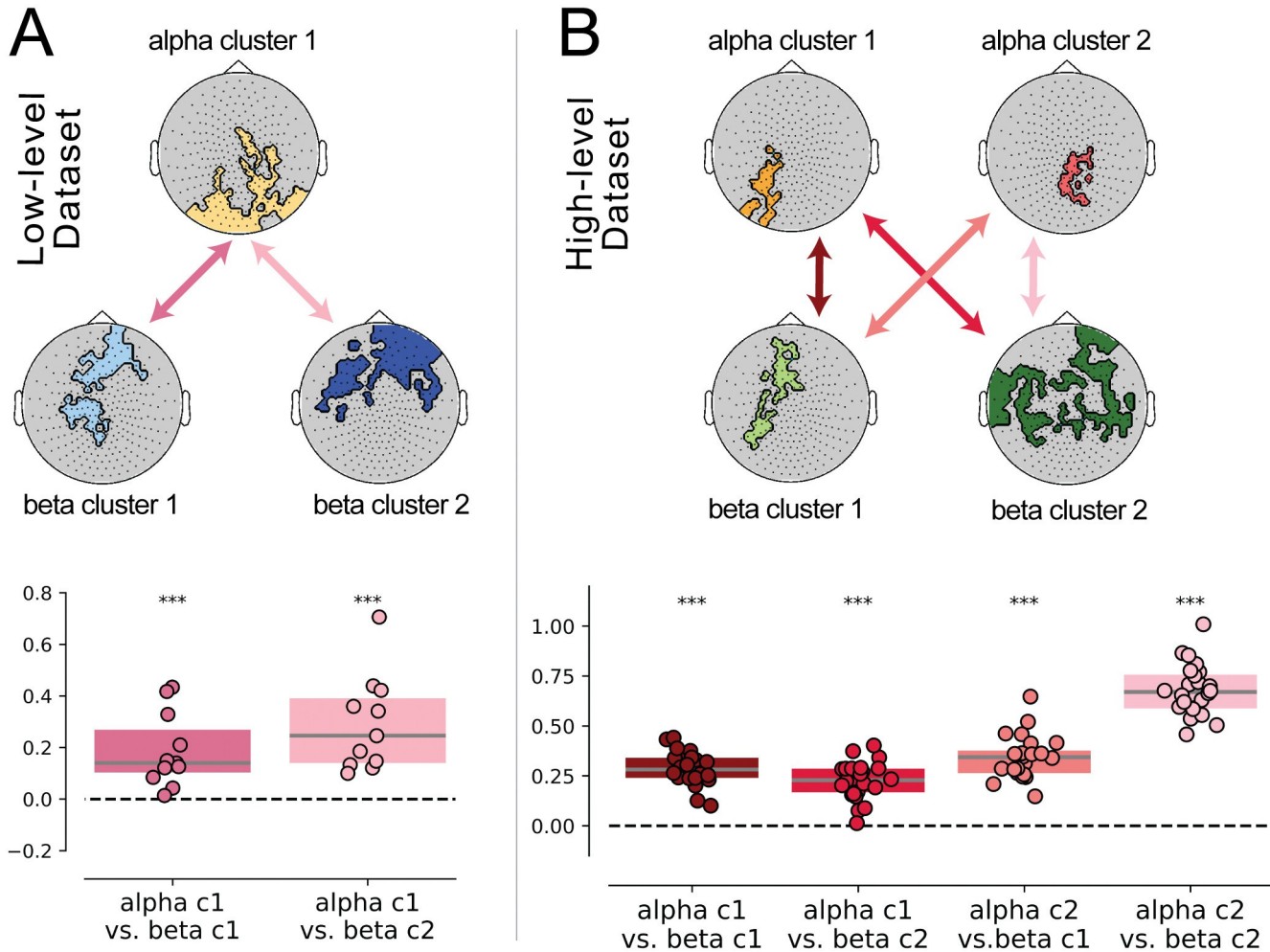

**Fig 5. Alpha power and beta power are positively correlated.** (**A**) Correlation between alpha power and beta power across single trials, between alpha cluster 1 and beta cluster 1 (left bar) and between alpha cluster 1 and beta cluster 1 (right bar) of the low-level data set. (**B**) Correlation between alpha power and beta power across single trials, between all pairs of alpha-beta clusters of the high-level dataset. In both A and B, boxplots represent z-transformed correlation coefficients across subjects and were tested against 0 with a Wilcoxon signed-rank test. For all boxplots, the solid line represents the median and the edges indicate the quartiles; * $p < 0.05$, ** $p \leq 0.01$, *** $p \leq 0.001$. The data underlying this figure can be found as Fig 5_Data at https://doi.org/10.5281/zenodo.14291607.

than the chance level of 0.5 for alpha power ($p = 0.0034$) but not for its residuals ($p = 0.087$), suggesting that some of the variance in perceptual outcome explained by alpha power is lost when beta power is regressed out, even though the differences were not statistically significant. Further, for both beta clusters, we found that the AUROC scores for beta power and beta power residuals after removing shared variance with the alpha cluster had marginally significant differences (beta cluster 1: $p = 0.071$, $W = 1.81$; beta cluster 2: $p = 0.053$, $W = 1.94$). The AUROC scores were not significant for beta power residuals (beta cluster 1: $p = 0.83$; beta cluster 2: $p = 0.92$, $W = 1.94$). These results suggest that in the low-level data set, alpha and beta power had significantly shared variance in their influences on perception.

In the high-level data set (**Fig 5B**), scores for alpha power and alpha power residuals after removing variance from both beta clusters did not significantly differ in alpha cluster 1 ($p = 0.21$, $W = 1.26$) but did in alpha cluster 2 ($p = 0.027$, $W = 2.21$). The AUROC scores were significantly higher than a chance level of 0.5 for alpha power and alpha power residuals

($p < 0.009$). Further, in both beta clusters, scores for beta power and beta power residuals after removing variance from both alpha clusters did not significantly differ (beta cluster 1: $p = 0.35$, $W = 0.93$; beta cluster 2: $p = 0.38$, $W = 0.89$). The AUROC scores were all significantly higher than a chance level of 0.5 ($p < 0.001$). These results suggest that in the high-level data set, alpha and beta power had both shared and independent influences on perception. That is, after removing their shared variance, both alpha power and beta power still individually predicted perception.

Together, these results suggest that, unlike the relationship between oscillatory power and SCP, alpha power and beta power did have partially shared influences on perception.

## Oscillatory power and SCPs correlate with pupil size

Arousal fluctuates constantly within the normal wakefulness state and can be tracked by the moment-to-moment changes in pupil diameter [39,51]. Spontaneous fluctuations of pupil size track the firing rates of locus coeruleus (LC) neurons, part of the ascending arousal system, which send widespread noradrenergic projections to the cortex [52]. Previous studies have shown that pre-stimulus baseline pupil size influences perceptual decision-making, including both reaction times and discrimination accuracy [30,53]. Using the high-level data set investigated herein, we previously reported that pupil size in a 2-s pre-stimulus window predicts subjective recognition of object images from trial to trial and modulates widespread cortical activity power measured by MEG [37]. These observations motivated us to hypothesize that the present findings, showing independent influences of oscillations and SCPs on perception, could be better understood by considering their respective relationships with pupil-linked arousal.

To investigate this question, we utilized the high-level data set, which had concurrent pupillometry and MEG recordings. For each pre-stimulus neural activity found to influence perception, we tested whether it was significantly correlated with pre-stimulus pupil size from trial to trial. Using the alpha clusters identified in the earlier analysis (**Fig 2E**), we found that alpha power from both sensor clusters were positively correlated with pupil size (Wilcoxon signed-rank test: $p = 0.023$, $W = 64$, $BF_{10} = 4.114$; $p = 0.012$, $W = 57$, $BF_{10} = 5.416$, respectively; **Fig 6A**). Similarly, we found that beta power in both sensory clusters (**Fig 3C**) positively correlated with pupil size ($p = 0.0014$, $W = 38$, $BF_{10} = 25.189$; $p < 0.001$, $W = 28$, $BF_{10} = 171.046$, respectively; **Fig 6B**). Finally, the SCP d.v. was also significantly correlated with pupil size from trial to trial ($p < 0.001$, $W = 30$, $BF_{10} = 63.066$; **Fig 6C**).

Since both alpha and beta power, as well as SCP d.v., were higher in the pre-stimulus period in recognized than unrecognized trials in the high-level data set (**Figs 2D, 2E** and **3C**), their positive correlations with pupil size were consistent with an overall positive relationship between pupil size and recognition rate [37]. These results raise the question of the specific relationship between neural activity and pupil size's influences on perception, which we probe in detail below.

## Oscillatory power and SCP's influences on perception have distinct relationships to pupil size

Given that all 3 neurophysiological signals were correlated with pupil size, we next investigated the relationship between their perceptual influences and the influence of pupil-linked arousal on perception. Specifically, are oscillatory power/SCP's influences on conscious perception mediated by pupil-linked arousal, or, alternatively, are they the mediators of pupil-linked arousal's influences on perception? To answer this question, we conducted 3 sets of mediation

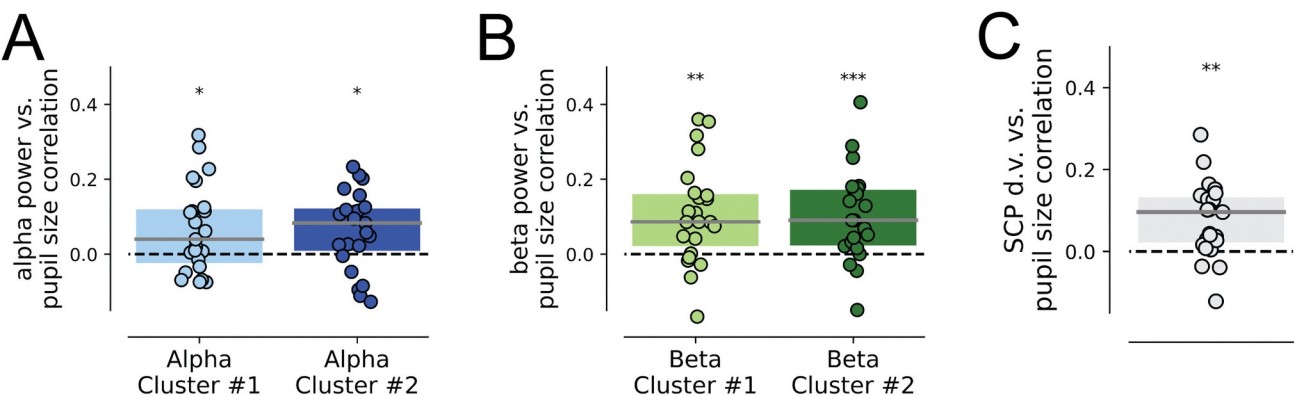

**Fig 6. Alpha and beta oscillations, as well as SCP d.v., positively correlate with pupil size.** (**A**) Correlation between alpha power and pupil size across single trials, for both alpha power clusters of the high-level data set. (**B**) Correlation between beta power and pupil size across single trials, for both beta power clusters of the high-level data set. (**C**) Correlation between SCP d.v. and pupil size across the entire pre-stimulus interval in the high-level data set. In A–C, boxplots represent z-transformed correlation coefficients across subjects and were tested against 0 with a Wilcoxon signed-rank test. Dashed horizontal lines represent a null correlation. The solid line represents the median and the edges indicate the quartiles. * $p < 0.05$, ** $p \leq 0.01$, *** $p \leq 0.001$. The data underlying this figure can be found as Fig 6_Data at https://doi.org/10.5281/zenodo.14291607. d.v., decision variable; SCP, slow cortical potential.

analyses (for details, see Methods) to test the respective relationship of alpha power, beta power, and SCP d.v. with pupil size in their perceptual influences.

The first set of mediation analyses concerned the relationship between pre-stimulus alpha power, pre-stimulus pupil size, and recognition rate (**Fig 7A**). We first tested whether pre-stimulus alpha power's influence on the recognition rate was mediated by pupil size, in each of the alpha clusters (shown in **Fig 2E**). To do so, we first tested the basic conditions of mediation: we assessed the direct effect between pre-stimulus alpha power and pupil size (**a**), the direct effect between pre-stimulus alpha power and recognition rate (**d**), and the indirect effect between pupil size and recognition rate, when pre-stimulus alpha power was controlled for (**e**). In the first alpha cluster, we found that coefficients **a** (one-sample $t$ test: $t_{22} = 3.30$, $p = 0.0033$), **d** ($t_{22} = 2.98$, $p = 0.007$), and **e** ($t_{22} = 2.30$, $p = 0.031$) were all significantly positive. Since these conditions were met, to find evidence of mediation, the indirect effect between pre-stimulus alpha power and recognition rate, when controlling for pupil size (**f**), must be smaller than the direct effect between pre-stimulus alpha power and recognition rate (**d**). We thus assessed whether d > f and found that the difference was not significant ($t_{22} = -2.69$, $p = 0.99$, $BF_{10} = 0.071$). We tested the same model in the second alpha cluster and found that all 3 basic conditions were met ($t_{22} = 3.27$, $p = 0.003$; $t_{22} = 3.06$, $p = 0.006$; $t_{22} = 2.91$, $p = 0.008$ for **a**, **d**, and **e**, respectively), but again the mediation effect was not significant ($t_{22} = -2.3$, $p = 0.98$, $BF_{10} = 0.078$). Therefore, we found no evidence of mediation between pre-stimulus alpha power and recognition rate via pupil size, in either alpha cluster (model indicated by the gray box in **Fig 7A**).

We then tested the alternative mediation model, whereby pupil size's influence on recognition rate is mediated by pre-stimulus alpha power. To do so, we first tested the basic conditions of mediation: we checked that the direct effect between pupil size and pre-stimulus alpha power (**b**), and between pupil size and recognition rate (**c**) and the indirect effect of pre-stimulus alpha power on recognition rate, controlling for pupil size (**f**) were significant. We found that all 3 conditions were met in the first alpha cluster ($t_{22} = 3.30$, $p = 0.003$; $t_{22} = 4.76$, $p < 0.001$; $t_{22} = 4.75$, $p < 0.001$ for **b**, **c**, and **f**, respectively) and in the second alpha cluster ($t_{22} = 3.27$, $p = 0.003$; $t_{22} = 4.86$, $p < 0.001$; $t_{22} = 4.77$, $p < 0.001$ for **b**, **c**, and **f**, respectively). We then tested the mediation effect by assessing whether the indirect effect between pupil size and

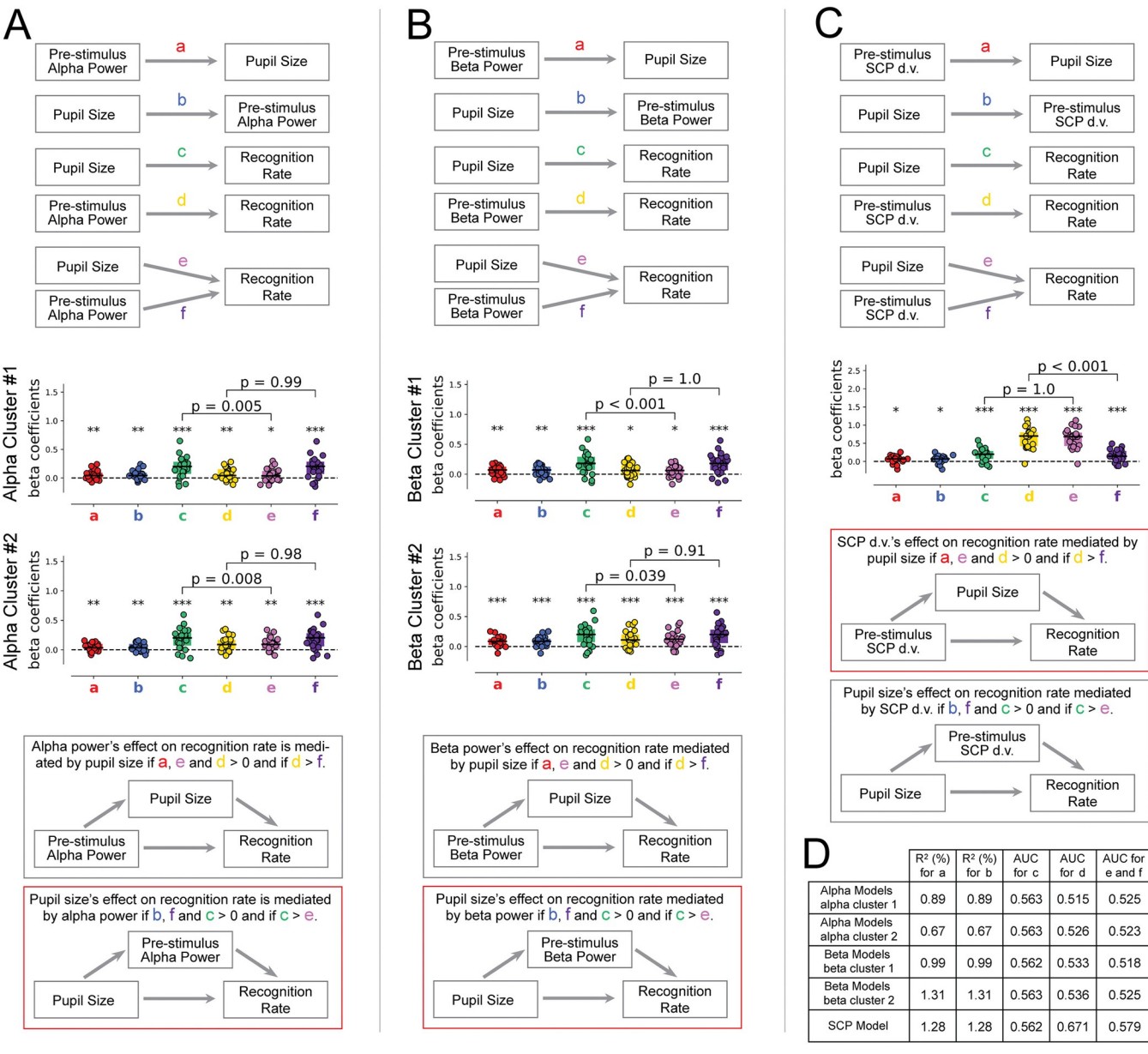

**Fig 7. Alpha power, beta power, and SCP d.v. are related to pupil size, which is thought to reflect arousal.** (**A**) The first set of mediation analyses explored the relationship between pre-stimulus alpha power, pupil size, and recognition rate. Model coefficients for the direct and indirect effects depicted in the top panel were computed and are plotted below for each alpha cluster. All initial conditions (see bottom panel) for both mediation models were met, but only the mediation effect for the model in which pupil size influences recognition rate via pre-stimulus alpha power was significant, as depicted by the red box. (**B**) The second set of mediation analyses explored the relationship between pre-stimulus beta power, pupil size, and recognition rate. Model coefficients for the direct and indirect effects depicted in the top panel were computed and are plotted below for each beta cluster. All initial conditions (see bottom panel) for both mediation models were met, but only the mediation effect for the model in which pupil size influences recognition rate via pre-stimulus beta power was significant, as depicted by the red box. (**C**) The third set of mediation analyses explored the relationship between pre-stimulus SCP d.v., pupil size, and recognition rate. Model coefficients for the direct and indirect effects depicted in the top panel were computed and are represented below. All initial conditions (see bottom panel) for both mediation models were met, but only the mediation effect for the model in which SCP d.v. influences recognition rate via pupil size was significant, as depicted by the red box. For all mediation analyses in A–C, direct effects between the 3 variables were captured with linear regression coefficients for a and b and logistic regression coefficients for c and d. Indirect effects of both predictor variables and recognition rate were captured with multiple logistic regression coefficients e and f. All coefficients are standardized. Boxplots represent z-transformed correlation coefficients across subjects and were tested against 0 with a Wilcoxon signed-rank test. Dashed horizontal lines represent a null correlation. For all boxplots, the solid line represents the median and the edges indicate the quartiles. * $p < 0.05$, ** $p \leq 0.01$, *** $p \leq 0.001$. (**D**) Table containing metrics for assessing model fits for each of the direct and indirect effects computed for all mediation analyses. Model fits for all linear regression models are assessed with the $R^2$ value in percentage points. Model fits for all logistic regression models are assessed with the AUC. The data underlying this figure can be found as Fig 7_Data at https://doi.org/10.5281/zenodo.14291607. AUC, area under the curve; d.v., decision variable; SCP, slow cortical potential.

recognition rate, when controlling for pre-stimulus alpha power (**e**), was smaller than the direct effect between pupil size and recognition rate (**c**). We found significant evidence for mediation (i.e., **c** > **e**) in both alpha clusters ($t_{22}$ = 2.78, $p$ = 0.005, $BF_{10}$ = 7.29 for alpha cluster 1, $t_{22}$ = 2.63, $p$ = 0.008, $BF_{10}$ = 5.58 for alpha cluster 2). Therefore, the influence of pupil size on recognition rate is mediated by pre-stimulus alpha power, in both alpha clusters (red box, **Fig 7A**).

The second set of mediation analyses concerned the relationship between pre-stimulus beta power, pupil size, and recognition rate (**Fig 7B**). We first tested whether pre-stimulus beta power's influence on the recognition rate was mediated by pupil size, in each of the beta clusters (shown in **Fig 3C**). We found that the 3 basic conditions for mediation were met in the first cluster ($t_{22}$ = 3.58, $p$ = 0.002; $t_{22}$ = 2.74, $p$ = 0.012; $t_{22}$ = 2.52, $p$ = 0.02 for **a**, **d**, and **e**, respectively) and in the second cluster ($t_{22}$ = 4.60, $p$ < 0.001; $t_{22}$ = 4.40, $p$ < 0.001; $t_{22}$ = 3.99, $p$ < 0.001 for **a**, **d**, and **e**, respectively). We then tested for the possibility that beta power's influence on perception was mediated by pupil size (i.e., **d** > **f**) and found no evidence of mediation in either cluster ($t_{22}$ = −3.41, $p$ = 1.0, $BF_{10}$ = 0.062; $t_{22}$ = −1.42, $p$ = 0.91, $BF_{10}$ = 0.103).

We then tested the alternative mediation model, whereby pupil size's influence on recognition rate is mediated by pre-stimulus beta power. We also found the 3 basic conditions for mediation were met in the first cluster ($t_{22}$ = 3.58, $p$ = 0.002; $t_{22}$ = 4.86, $p$ < 0.001; $t_{22}$ = 4.88, $p$ < 0.001 for **b**, **c**, and **f**, respectively) and in the second cluster ($t_{22}$ = 4.60, $p$ < 0.001; $t_{22}$ = 4.86, $p$ < 0.001; $t_{22}$ = 4.75, $p$ < 0.001 for **b**, **c**, and **f**, respectively). We then tested for the possibility that pupil's influence on perception was mediated by beta power (i.e., **c** > **e**) and found significant evidence of mediation in both clusters ($t_{22}$ = 3.72, $p$ < 0.001, $BF_{10}$ = 43.38 for beta cluster 1, $t_{22}$ = 1.85, $p$ = 0.039, $BF_{10}$ = 1.60 for beta cluster 2). Therefore, the influence of pupil size on recognition rate is also mediated by pre-stimulus beta power, in both beta clusters (**Fig 7B**, red box).

The third set of mediation analyses concerned the relationship between pre-stimulus SCP d.v., pupil size, and recognition rate (**Fig 7C**). We first tested whether the influence of pre-stimulus SCP d.v. on recognition rate was mediated by pupil size. We found that the 3 basic conditions for mediation were met ($t_{22}$ = 2.72, $p$ = 0.013; $t_{22}$ = 11.01, $p$ < 0.001; $t_{22}$ = 10.93, $p$ < 0.001 for **a**, **d**, and **e**, respectively). We tested the mediation effect (i.e., **d** > **f**) and found significant evidence of mediation ($t_{22}$ = 3.72, $p$ < 0.001, $BF_{10}$ = 2.08e5). These results suggest that the influence of SCP d.v. on recognition rate is mediated by pupil size (**Fig 7C**, red box).

We then tested the alternative mediation model, whereby pupil size's influence on recognition rate is mediated by pre-stimulus SCP d.v. We also found that the 3 basic conditions for mediation were met ($t_{22}$ = 2.72, $p$ = 0.013; $t_{22}$ = 4.88, $p$ < 0.001; $t_{22}$ = 4.27, $p$ < 0.001 for **b**, **c**, and **f**, respectively). We tested the mediation effect (i.e., **c** > **e**) and found no evidence of mediation ($t_{22}$ = −7.41, $p$ = 1.0, $BF_{10}$ = 0.017).

In all analyses, the overall performance of each of the models was quantified using the coefficient of determination $R^2$ for linear regression models and using AUC (area under the ROC curve) for logistic regression models, as summarized in **Fig 7D**. Together, these findings suggest that oscillatory powers and SCPs have distinct relations to pupil-linked arousal in their influences on conscious perception. While SCP d.v.'s influence on conscious perception is mediated by pupil-linked arousal, pupil-linked arousal's perceptual influence is itself mediated by the wax-and-wane of alpha and beta oscillations (**Fig 7**, red outlines).

## Discussion

Spontaneous power fluctuations of brain oscillations and spontaneous variations in large-scale SCP activity patterns have both been found to predict conscious perception. Here we show, for

the first time, that their influences on perception act through independent channels. In addition, while SCP's perceptual influence was partially mediated by pupil-linked arousal, alpha and beta powers acted as partial mediators of pupil-linked arousal's influence on perception. These results establish that aperiodic brain activity and brain oscillations have independent influences on perception. Importantly, our main findings showing independence between pre-stimulus SCP and oscillation's effects on perception were reproduced in 2 separate data sets involving different visual stimuli, showing the reproducibility and generalizability of the present findings. In what follows, we discuss the implications of these findings.

In both data sets, conscious perceptual outcome, whether it was the detection of a Gabor patch or the recognition of a meaningful content, could be decoded from the whole-brain pattern of SCPs in the pre-stimulus interval [10,12]. SCPs correspond to fluctuations in brain field potentials comprising the low-frequency end (<5 Hz) of the aperiodic spectrum and reflect fluctuations in cortical excitability modulated by synaptic activity in the superficial layers [44,54]. Indeed, tasks presented on the negative shifts of spontaneous SCP fluctuations are solved faster [55,56], more accurately [57], and have a lower sensory threshold [34]. Further, negative SCPs in the pre-stimulus interval were shown to be associated with the awareness level of a visual stimulus [58]. Although the neural mechanism by which spontaneous SCPs modulates perceptual awareness is not yet fully understood, our previous study using the high-level data set showed that the SCP d.v. extracted using the present approach predicts conscious recognition in a "non-content-specific" manner [12], such that the same pre-stimulus activity pattern facilitates recognition of images from different categories. This observation is compatible with the present finding that SCP's influence on conscious perception is at least partially mediated by pupil-linked arousal mechanisms (**Fig 7C**). We note that this earlier study [12] also observed a "content-specific" process in the pre-stimulus SCPs, where a different activity pattern promotes the recognition of each object category, and this content-specific SCP activity was not correlated with pupil size. Our study did not probe the content-specific spontaneous SCP process, because whether there are similar processes in spontaneous alpha and beta power fluctuations that influence perception in a content-specific manner remains to be seen.

Concerning alpha oscillations, the most frequently documented effect on conscious perception is that pre-stimulus alpha power in occipital regions is inversely related to the probability of consciously perceiving a visual stimulus [6,13–17]. Here, in the low-level data set, we replicated this effect: alpha power in a group of occipital sensors was significantly lower preceding consciously detected stimuli (**Fig 2B**). In line with previous findings, this likely reflects excitability fluctuations in visual areas that influence how an incoming stimulus is processed. Supporting this interpretation, occipital alpha power predicts the probability of perceiving a phosphene elicited by a TMS pulse, a phenomenon known to vary as a function of visual cortex excitability [59,60]. Changes in excitability gated by alpha oscillations could reflect waves of inhibition in sensory regions [61,62], such that lower alpha power indicates a disinhibition that facilitates sensory processing.

In the high-level data set, we observed a different pattern, with higher alpha power preceding consciously recognized stimuli, accompanied by a distinct spatial topography centered in more anterior regions. This suggests a distinct mechanism through which alpha oscillations may influence visual perception. In the high-level data set, instead of detecting a visual signal from a uniform background where the excitability of sensory regions might play a stronger role (as in the low-level data set), subjects' task was to detect a meaningful content in a complex visual image which requires object segmentation. Here, top-down gating or feedback from higher-order areas likely play a more important role. Alpha oscillations have been implicated in attention- and prediction-mediated facilitation of specific visual representations [19,48–50,63–65]. For instance, pre-stimulus feedback connectivity in the alpha range can bias the

content of visual perception [66]. Additionally, shifts in attention can modulate parietal excitability via changes in alpha power [48]. Further, pre-stimulus alpha power in frontoparietal areas predicts individual perceptual biases [67,68]. These studies collectively support a role for frontoparietal alpha power in mediating top-down signals that regulate perception and are consistent with models suggesting a role for frontal alpha power in cognitive control [27].

Another, not mutually exclusive possibility is that the perceptually relevant alpha power we identified in the high-level data set might originate from the cingulo-opercular network that controls tonic alertness [27,28]. This interpretation is consistent with our finding that alpha power partially mediates pupil-linked arousal's influence on recognition rates. Importantly, spontaneous activation of the anterior insular cortex, a key node of the cingulo-opercular network, precedes spontaneous phasic pupil dilation [69], and these brain regions have been suggested to be part of the "central autonomic network" involved in controlling the autonomic nervous system which in turn controls spontaneous changes in pupil size [69].

Importantly, despite the notable differences in the specific relations between alpha oscillations and conscious perception in the 2 data sets, in both cases we found that alpha power and SCP d.v. were not correlated with each other, even though both signals were predictive of perceptual outcome. This suggests that these 2 measures of spontaneous activity likely have independent sources and mechanisms of action on perception. This conclusion is consistent with models suggesting that 1/f aperiodic activity and oscillations add on top of each other in both the time domain and frequency domain (the "additive model") and have separate sources [45,70].

Similarly, for beta oscillations, we found that beta power and conscious perception were inversely related in the low-level data set, and positively related in the high-level data set—again, with more anteriorly located clusters in the high-level data set. The underlying mechanisms of beta power's influences on visual perception are far less studied than for alpha oscillations. Beta oscillations have been implicated in perceptual expectations [71] and the reactivation of content-specific perceptual representations [72,73]. Additionally, an earlier study reported a causal role for beta power in the frontoparietal network in conscious visual perception [74]. Importantly, here too, despite different effects of beta power on conscious perception as a function of the visual task, we found no significant relationship between perceptually relevant measures of beta power and SCPs. Although we did find a positive correlation between beta power and SCP d.v. in the high-level data set, our follow-up analysis showed that each measure accounted for non-overlapping variance in the perceptual outcome, suggesting that they exerted independent influences on perception.

Finally, in the high-level data set which had concurrent MEG and pupil recordings, we found that pre-stimulus alpha power, beta power, and SCP d.v. all correlated with pre-stimulus pupil size across trials. This finding indicates that all 3 measures of spontaneous brain activity may be related to fast changes in arousal within the waking state [39,51], despite the mutual independence of their perceptual influences. We probed this issue further using a series of mediation analyses, which revealed distinct relationships of oscillatory power and SCP with regard to pupil size in their influences on perception.

We found that pre-stimulus alpha and beta power partially mediate pre-stimulus pupil size's influence on recognition (**Fig 7A and 7B**). For alpha power, this effect can be potentially explained by its relationship to tonic alertness, as discussed earlier. In addition, a recent study showed that noradrenergic activity—which is a central mechanism in controlling pupil size— is associated with changes in alpha power [75]. Beta power's effect could potentially be explained by its strong correlation with alpha power's influence on perception (**Fig 5**). Our observation of a positive correlation between alpha power and pupil size is consistent with a recent study [76]. This study, using a range of stimulus contrasts, revealed that alpha power

and pupil size had, respectively, an additive versus a multiplicative influence on detection across stimulus contrast levels [76]. Our finding showing that alpha power partially mediated pupil's influence on perception is not incompatible with this earlier study and might help to explain earlier findings showing that baseline pupil size can influence both the sensitivity and criterion of conscious detection [37].

We also found that SCP's influence on conscious recognition was at least partly mediated by pupil size. This suggests that the large-scale SCP activity patterns modulate the pupil-linked arousal system, which may in turn influence conscious perception. SCPs may influence the activation of the locus coeruleus-norepinephrine (LC-NE) system through top-down cortical projections, thereby influencing pupil size. In turn, shifts in pupil-linked arousal can influence visual perception through a variety of neural mechanisms, including the release of NE and acetylcholine in the cortex [52,77,78], which in the visual cortex could enhance the signal-to-noise ratio during visual processing [38]. In addition, pupil-linked arousal changes could enhance thalamic function [79,80] such that visual stimuli are more readily processed. Further research is needed to establish the precise neural mechanisms governing the complex interplay between brain oscillations, SCPs, physiological arousal, and visual awareness.

In summary, our results, obtained using 2 data sets involving different visual tasks and stimuli, demonstrate that spontaneous aperiodic fluctuations and oscillatory activity influence conscious perception through independent mechanisms. They also highlight the complex relationships between spontaneous neural activity and arousal in modulating conscious perception. These results shed new light on our understanding of how visual awareness arises through intricate interactions between spontaneous ongoing brain activity and incoming sensory inputs and pave the way for future dissection of these mechanisms through causal manipulations or circuit-level studies.

## Methods

### Subjects

This study utilized 2 data sets previously collected in our laboratory that were published as [10] and [12]. We will refer to these datasets as the "low-level data set" and the "high-level data set," respectively.

Both sets of low-level data set subjects ($N = 11$, 6 females, mean age 27, age range 22 to 38) and high-level data set subjects ($N = 25$, 15 females, mean age 26, age range 22 to 34) were right-handed, neurologically healthy, and had normal or corrected-to-normal vision. All subjects provided written informed consent. Both experiments were approved by the Institutional Review Board of the National Institute of Neurological Disorders and Stroke (protocol #14-N-0002). In the high-level data set study, one enrolled participant decided to stop the experiment after finishing one experiment block due to discomfort and was not included in data analyses.

### Stimuli and task

**Low-level data set.** Stimuli were presented on a Panasonic DLP projector with a 60-Hz refresh rate, onto a screen 75 cm away from the subject's eyes. An optical filter was placed on the lens to reduce the luminance of the stimulus so that every subject could reach a subjective threshold of stimulus duration longer than 16.7 ms ± the limitation of the projector refresh rate. All subjects were dark adapted for at least 30 min before the experiment began.

In each trial, a white fixation cross was first presented on a gray background (**Fig 1A**) and the trial began when the subject pressed a button to indicate they were ready to begin. A blank screen appeared for a random duration between 2 and 6 s, with the duration following an exponential distribution across trials to reduce expectation effects. Then, a Gabor patch (1±

visual angle/cycle; 1% contrast) was presented for a short duration, with the exact duration set individually to control for subjective perception rates; see below. The orientation of the Gabor was either 45˚ or 135˚ with equal probability. The background luminance of the stimulus screen was equal to the luminance of the blank screen. Six to 10 trials per subject were catch trials in which no stimulus was presented. Subjects were not instructed that catch trials would be included. Another blank screen was then presented for 3 to 6 s, again following an exponential distribution. Three questions were presented at the end of each trial: (i) a two-alternative forced choice: was the Gabor pointing to upper-left (135˚) or upper-right (45˚)? (ii) Did you see the Gabor patch or not? (iii) On a scale of 1 to 4, how confident are you about your response to question ii? Subjects responded via a fibreoptic keypad (Lumi-Touch). Only the responses to question ii were used for the neural analyses of the current study.

Each experimental session began with a first stage in which the duration of the Gabor patch was adjusted with Levitt's staircase method, which was used to individually titrate the threshold for subjective awareness (i.e., the subject responds "seen" to question ii in about 50% of trials). Durations across subjects ranged 33.3 ms (6 subjects), 50 ms (3 subjects), and 66.7 ms (2 subjects).

In a second stage of the experiment, subjects were shown trials with stimuli at their threshold duration while MEG signals were continuously recorded. Sessions of trials were less than 12-min long and subjects were allowed to rest between sessions. In total, each subject performed 113 to 277 trials (mean ± SD across subjects: 183.5 ± 56.4) after artifact rejection. The entire experiment, including dark adaptation and staircase, lasted up to 3 h per subject. The present data were previously reported in [10] and [42]. Full details of the task paradigm are described therein.

**High-level data set.** Stimuli consisted of images selected from 4 categories: faces, animals, houses, and objects (see **Fig 1C**). The images were selected from public domain labeled photographs from the Psychological Image Collection at Sterling (PICS, http://pics.psych.stir.ac.uk/) and were resized to $300 \times 300$ pixels and converted to grayscale. The pixel intensities ranged from 0 (black) to 255 (white), were normalized by removing the mean and dividing by the standard deviation, and were filtered using a 2D Gaussian smoothing kernel with a standard deviation of 1.5 pixels and $7 \times 7$ pixels size (imgaussfilt, MATLAB). Five unique images were included in each category, resulting in 20 total images. Scrambled images were created by shuffling the 2D Fourier transformed phase of one randomly chosen image from each category. The edges of the images were gradually brought to background intensity by multiplying the image intensity with a Gaussian window with a standard deviation of 0.2.

Stimuli were presented via a projector with a 60-Hz refresh rate onto a screen, using the Psychophysics Toolbox64 in MATLAB. Stimuli were 8˚ in diameter and presented gradually from 0.01 to threshold intensity for approximately 66.7 ms (i.e., 4 video frames).

In each trial, a gray background was first shown with a central fixation cross for a random duration between 3 to 6 s, with the duration following an exponential distribution. A stimulus was then shown and followed by another blank screen presented for a random duration between 2 and 4 s (again following an exponential distribution). The luminance of the gray blank screens was equal to the background luminance of the stimulus screen. The first blank period insured that the onset of the stimulus was not predictable. Two questions were presented at the end of each trial. (i) A four-alternative forced choice: was the image category of the stimulus a face, house, object, or animal? If the image was not recognized, subjects were instructed to make a random guess, with an emphasis on making a genuine guess to the best of their ability. The response mapping indicated by the order of the 4 category words on the screen was randomized across trials. (ii) Did you recognize the stimulus, yes or no? "Yes" was defined as "something that makes sense in the real world," and subjects were instructed to respond "yes" even if the image was not entirely clear, or they saw only part of it (for example, if

they saw only the eyes and ears of an animal without knowing exactly what type of animal it was). Subjects were instructed to respond "no" if they did not see anything at all or if they saw random noise patterns. Subjects were not informed of the use of scrambled images. Subjects responded via a right-hand keypad. A central fixation cross was presented at all times except during response prompts and subjects were instructed to fixate on the cross at all times it was present.

Each experimental session began with an adaptive staircase procedure "Quest" [81] to titrate the image contrast to yield a recognition rate of 50% (proportion of "yes" responses to question ii). The image pixel intensity $I$, at a given contrast $c$, was calculated as follows:

$$I(c) = b(I_{scaled}*c + 1),$$

where $b$ is the background intensity set to a constant value of 127 and scaled pixel intensities $I_{scaled}$ were obtained by rescaling the image pixel intensities between −1 and 1. Therefore, the lightest pixel value in the image was equal to $I_{max} = b(1+c)$ and the darkest $I_{min} = b(1-c)$. Therefore, the contrast of a presented image was defined as follows:

$$c = \frac{I_{max} - I_{min}}{2b},$$

which ranged between 0 and 1.

Subjects performed the Quest procedure under the same conditions as in the main task, except that trial timing was faster (750 ms pre-stimulus interval and 1 s post-stimulus interval). The initial contrast for each image was determined by the mean threshold contrasts obtained in 3 pilot subjects who were not included in the present study. The Quest procedure included 120 trials, split into 3 individual staircase procedures, and the median of the 3 threshold contrasts was used for the main task.

After the Quest procedure, the main experiment began, in which trials were shown with the determined stimulus contrast while MEG signals were continuously recorded. The experiment had 360 trials with 300 real-image trials and 60 scrambled-image trials. Each unique image was repeated 15 times and each image was presented in a random order, such that image category was unpredictable. The trials were split into 10 blocks of 36 trials each, and each block ended with a self-paced break period.

## Data acquisition

**Low-level data set.** Experiments were conducted in a whole-head 275-channel CTF MEG scanner (VSM MedTech). Two dysfunctional sensors were removed from all analyses. MEG data were collected with a sampling rate of 600 Hz with an anti-aliasing filter at <150 Hz. Before and after each recording session, the subject's head position was measured using coils placed on the nasion and the left and right preauricular points. All MEG data samples were corrected with respect to the refresh delay of the projector (measured with a photodiode).

**High-level data set.** Experiments were conducted in a whole-head 275-channel CTF MEG scanner (VSM MedTech). Three dysfunctional sensors were removed from all analyses. MEG data were collected with a sampling rate of 1,200 Hz. The head position of the subject was measured before and after each block using coils placed on the ear canals and the bridges of the nose. Between blocks, the head position of the subject was measured with respect to the MEG sensory array using coils placed on the nasion and the left and right preauricular points. The subject self-corrected their head position to the same position recorded at the start of the first block using a custom visual-feedback program inspired by Stolk and colleagues [82], in order to minimize head displacement across the experiment. All MEG data samples were corrected with respect to the refresh delay of the projector (measured with a photodiode).

For this data set, we also recorded eye-tracking data, whereby we recorded subjects' eye position and pupil size continuously throughout the experiment using an Eyelink 1000+ system in the binocular mode, with a sample rate of 1,000 Hz.

## Data analysis

**Behavior.** *Low-level data set.*

Subjects were presented with left- or right-tilting Gabor patches, which were titrated to their individual detection thresholds. We computed the percentage of trials in which subjects detected a stimulus, in the stimulus-present trials, and in the catch trials. We also computed the percentage of trials in which subjects correctly discriminated the tilt of the stimuli, both in "seen" and "unseen" trials (i.e., based on responses to the detection question). See **Fig 1B**.

*High-level data set.*

Subjects were presented with object stimuli at the threshold of subjective recognition. We computed the percentage of trials in which subjects reported recognizing an object, in both real and scrambled trials. We also computed the percentage of trials in which subjects correctly categorized stimuli, in both "recognized" and "unrecognized" trials (i.e., based on responses to the recognition question) and for both real and scrambled images. See **Fig 1D**.

**Preprocessing.** *Low-level data set.*

MEG data was preprocessed using FieldTrip (http://fieldtrip.fcdonders.nl) in MATLAB (MathWorks). Each recording session was demeaned, detrended, band-pass filtered between 0.05 and 150 Hz with a fourth-order Butterworth filter, and notch-filtered at 60 and 120 Hz to remove power-line noise. Independent component analysis (ICA) was performed on the continuous data to remove eye-movement, cardiac, and movement artifacts. No frequency-domain filtering was applied in order to avoid artifactual signal bleeding from the post-stimulus signal into the pre-stimulus period. Subsequently, data were epoched from 2 s before to 3 s after stimulus onset and remaining trials with artifacts were rejected manually.

*High-level data set.*

We conducted the following preprocessing steps on our eye-tracking data. We identified blinks by finding time points where the diameter of the right eye pupil dropped by a threshold of 3.6 measurement units, with blink onset and offset defined as 40 ms before and after crossing the threshold, respectively. Time points affected by blinks were removed from further analyses. We averaged pupil diameter measurements within four 400 ms pre-stimulus intervals (−1.7 to −1.3 s, −1.3 to −0.9 s, −0.9 to −0.5 s, −0.5 to −0.1 s). No frequency domain filtering was applied. One subject was excluded from pupil size analysis due to noisy data and hence all analyses involving pupil size include 23 subjects instead of 24 (see **Figs 5** and **6**).

MEG data were preprocessed using Python and the MNE toolbox (version 0.17.1) [83]. ICA was performed on each block of the MEG data to remove eye-movement, blink, cardiac and movement-related artifacts. The linear trend was removed from each experimental block. No frequency-domain filtering was applied to avoid artifactual signal bleeding from the post-stimulus signal into the pre-stimulus period. Data were then epoched from 2 s to 3 s after stimulus onset. Based on our blink identification method for the eye-tracking data, we then excluded any trials in which a blink occurred during stimulus presentation, resulting in $12.0 \pm 2.3$ (mean ± SEM) rejected trials for 20 participants. Loss of eye-tracking occurred in 4 subjects in several experiment blocks; hence, no trials were rejected in those subjects based on blinks.

**Time-frequency analysis.** To obtain power changes in the alpha and beta frequency bands related to the detection (low-level data set) or recognition (high-level data set) of stimuli, we applied a wavelet transform (Morlet wavelets, 47 frequencies, frequency range: 0.8 to 40 Hz, number of cycles increasing linearly from 3 to 9, time window: −1.7 to −0.1 s relative to

stimulus onset) to the MEG timecourses. This time window was chosen to circumvent any edge effects as well as any temporal smearing due to wavelet convolution. Temporal smearing refers to the spreading of information from one time point to adjacent time points in the wavelet-transformed signal. This smearing could cause signal from the post-stimulus period to leak into the pre-stimulus power spectra due to the way wavelet analysis fits the data with wavelet functions across time. In Fig A in **S1 Text**, we show the results of a simulation in which we quantified the extent of temporal smearing in the pre-stimulus interval caused by a sinusoidal wave of 10 Hz, starting at time point 0. We applied a wavelet transform with the same parameters as above, and determined the extent of contamination by computing twice the standard deviation of the Gaussian envelope [84] and plot the limit before which data is guaranteed to not be contaminated by temporal smearing (red line in Fig A in **S1 Text**). Our frequency bands of interest, alpha and beta (represented in **Fig A** in **S1 Text** by green and blue hatchings, respectively), are not affected by any temporal smearing if pre-stimulus estimates are taken up to −0.1 s, hence our decision to use this time as our cut-off for the pre-stimulus interval.

We thus obtained a power spectrum for each trial, sensor, and 100 ms window (from −1.7 s to −0.1 s relative to stimulus onset). We then averaged power spectra across 400 ms windows to obtain a power spectrum for each trial, sensor, and 400 ms pre-stimulus window (4 windows from −1.7 to −1.3 s, −1.3 to −0.9 s, −0.9 to −0.5 s and −0.5 to −0.1 s). We then parametrized the power spectra into periodic and aperiodic signals using the *fooof* toolbox [47]. *fooof* takes the original power spectrum as input, computes an initial aperiodic fit, subtracts the aperiodic fit from the original power spectrum to obtain a flattened spectrum, and fits Gaussians to detect any oscillatory peaks remaining in the flattened spectrum. Oscillatory peaks are defined as any positive fluctuation in the flattened spectrum that is at least 1 standard deviation above the mean of the entire spectrum. With these parameters, we aimed to obtain oscillatory estimates in as many trials as possible, without assuming that we would find an oscillation in every trial, given the wide inter-trial variability of alpha and beta oscillations [85]. We obtained alpha oscillation estimates in 88% of trials, and beta oscillation estimates in 79.4% of trials. In a control analysis, we compared behavioral measures, SCP d.v. and beta power between trials with an alpha peak and those without an alpha peak and found no significant differences. We ran the same comparison for beta peaks and also found no significant differences. These results show that this selection of trials was representative of the entire sample in terms of behavioral and neural metrics.

We therefore obtained the center frequency (CF), amplitude, and bandwidth (BW) of any remaining oscillations after the aperiodic 1/f signal had been accounted for. For each trial, we obtained measures for alpha oscillations if any oscillatory peak was present in the 7 to 14 Hz band, and for beta oscillations, if any oscillatory peak was present in the 14 to 30 Hz band. For both, if more than one oscillation was found in the frequency band of interest, the oscillation with the highest amplitude was chosen.

To obtain the power of alpha and beta oscillations, we computed the area under the curve (AUC) between one bandwidth ($BW_\alpha$ or $BW_\beta$) below the center frequency of the detected oscillation ($CF_\alpha$ or $CF_\beta$) to one bandwidth above it, as follows:

$$AUC_\alpha = \sum_{CF_\alpha - BW_\alpha}^{CF_\alpha + BW_\alpha} total - \sum_{CF_\alpha - BW_\alpha}^{CF_\alpha + BW_\alpha} aperiodic$$

$$AUC_\beta = \sum_{CF_\beta - BW_\beta}^{CF_\beta + BW_\beta} total - \sum_{CF_\beta - BW_\beta}^{CF_\beta + BW_\beta} aperiodic$$

For both equations, *total* is the total linear spectrum, including both aperiodic fit and oscillations, and *aperiodic* is the aperiodic spectrum. See **Fig 1F** (left panel) for illustration.

We note that we used a frequency range of 2 bandwidths in order to capture the entire oscillatory peak, given that the bandwidth output by *fooof* is the bandwidth at full-width half-maximum, and thus captures only the upper half of the peak. See figure illustration in Note A in **S1 Text**.

The entire processing pipeline for extracting oscillatory power separately from the 1/f power spectrum is explained in detail in Note A in **S1 Text**, including step-by-step illustrations.

**Clustering analysis.** To test for statistically significant power differences between seen and unseen trials (low-level data set; includes all real and catch trials) or recognized and unrecognized trials (high-level data set; includes all real and scrambled trials), we used a cluster-based nonparametric randomization approach [86]. We first statistically compared averaged alpha power between sets of perceptual outcomes across all subjects, using the Wilcoxon signed-rank test. This was done independently for each sensor and time interval (four 400 ms pre-stimulus intervals described above). For group-level statistics, we analyzed the consistency of Wilcoxon values over subjects by means of a nonparametric randomization test identifying spatial clusters showing the same effect. Neighboring channels were defined based on spatial adjacency. Spatially adjacent W-values with the same sign exceeding an a priori-defined threshold ($p < 0.05$) were combined into a cluster. W-values were summed across sensors in the cluster to yield that cluster's summary statistic. We next computed a "null" distribution by randomly permuting the data, such that the data was randomly assigned to either of the 2 perceptual outcome labels, and statistical differences comparing the 2 perceptual outcomes were again computed using the Wilcoxon signed-rank test and spatially clustered, and each permuted cluster's summary statistic was obtained. This random assignment process was repeated 1,000 times, resulting in a W-value summary statistic for each repetition. The final cluster-corrected *p*-value for each cluster was calculated as the proportion of clusters from the permuted "null" distribution with a summary statistic larger than the absolute value of the summary statistic of the unpermuted cluster of interest. Clusters from the original unpermuted data were considered significant if their summary statistic exceeded the 95th percentile of the null distribution. See **Fig 1F** (right panel) for illustration.

For all further analyses, we averaged oscillatory (alpha or beta) power within significant clusters (defined as the time interval and the sensors at which there are statistically significant differences in power related to the detection/recognition of visual stimuli, as described above), for each trial and subject. We henceforth refer to these as our "perceptually relevant" measures of oscillatory power.

To compare the analysis pipelines between the low-level and the high-level data sets up to this point, we point out that all pre-processing and analysis steps were identical, with the only exception being that the high-level MEG data were further cleaned by rejecting trials based on eye-tracking data, which was collected for this data set, but was not available for the low-level data set. Further, the decoding analyses detailed below rely on different classifiers (SVM for low-level, logistic regression for high-level) to follow the approaches already utilized in the published studies that used these data. Otherwise, all analysis steps were identical between the 2 data sets.

**Decoding analysis.** *Low-level data set.*

For each subject, single-trial classification of "seen" versus "unseen" was performed using activity from all sensors using all real and catch trials. To build upon a previous study published with these data [10], we used the LIBSVM package [87] to implement a support vector machine (SVM), applied to pre-stimulus data downsampled to 10 Hz. A 5-fold cross-

validation scheme was applied, using 5 interleaved sets of trials. Trials were balanced in the training set by using a random subset of trials in which the number of trials was equalized between the 2 conditions. Classification was performed 10 times, each time using a different random subset of balanced trials for training, and performance was averaged across iterations. We used a cost parameter C = 2^-6 for all decoding analyses. We obtained the decision variable for each trial, which is a measure of each trial's distance from the decision hyperplane in high-dimensional space (see **Fig 1E**) and represents the evidence in favor of a specific perceptual outcome in that trial, based on activity across all sensors. The SCP decision variable (SCP d.v.) for each trial was averaged across the 10 classification repetitions. We thus obtained estimates for every 100 ms interval of the pre-stimulus period. To match the temporal resolution of our oscillatory estimates, we then averaged SCP d.v. across four 100 ms estimates to obtain values for 400 ms intervals.

*High-level data set.*

For each subject, single-trial classification of "recognized" versus "unrecognized" was performed using activity from all sensors using all real and scrambled trials. Multivariate pattern analyses were performed with scikit-learn package for Python [88] (version 0.20.2). Following a previous study published with these data [12], we used a penalized logistic regression model with L2 norm regularization (C = 1) and a leave-one-out cross-validation scheme. We used a Coordinate Descent (CD) algorithm to fit the binary model (i.e., predicting the recognition report: "yes"/"no").

The probability of forthcoming recognition experience was calculated for each trial using the logistic regression model:

$$\Pr(Y = y|x) = \frac{1}{1 + e^{-\beta_y x}},$$

where x is the pre-stimulus brain state, y is the predicted class label ("yes"/"no" for recognition), and $\beta_y$ is the fitted model parameters. Pre-stimulus brain state, x, was calculated using MEG data recorded from M sensors and down sampled to 10 Hz. We obtained estimates for every 100 ms interval and averaged them across time in four 400 ms windows (−1.7 to −1.3 s, −1.3 to −0.9 s, −0.9 to −0.5 s, −0.5 to −0.1 s) before stimulus onset, and, for every experimental trial, x constituted an M-dimensional vector. Each MEG sensor constitutes a model feature in this scenario, and $\beta_y$ is an M-dimensional vector of weights. All sensors were used to fit the model. We implemented leave-one-out cross-validation decoding across all real and scrambled trials. All 24 subjects were included in decoding analyses.

As a measure of decision variable, we used the probability value computed by our logistic regression classification for each trial. However, to correct for the non-normal distribution of the probability values, we applied a logit transform to all values and used these as our measure of SCP d.v. for all further analyses.

For both low-level and high-level datasets, we consider the SCP d.v. our "perceptually relevant" measure of SCPs. This measure quantifies the strength of pre-stimulus SCP signal that predicts the upcoming outcome of conscious perception.

**Correlation analysis.** To assess whether oscillatory power and SCP independently or interdependently influence perceptual outcome, we correlated both "perceptually relevant" measures of oscillatory and aperiodic activity across trials at the single-subject level. That is, we first correlated alpha power and SCP d.v. within each alpha power cluster; we then correlated beta power and SCP d.v. within each beta power cluster. Note that for plotting (scatter plots in **Figs 2 and 3**), but not analysis, alpha power and beta power were min-max normalized within subjects. We also correlated alpha power and beta power between all significant clusters. We used Spearman's rank correlation to correlate all measures across trials within each subject,

then assessed group-level statistics by testing the z-transformed rho values against zero with a Wilcoxon signed-rank test. We also performed one-sample Bayesian *t* tests using JASP [89], employing a Cauchy prior width of 0.707.

**AUROC analysis.** To assess whether any significant positive correlations between SCP d. v. and beta power were indicative of a shared influence on perceptual outcome, we built 2 linear models describing the linear relationship between SCP d.v. and beta power, as detailed below:

$$SCP_{predicted} = i + k*beta_{total} + SCP_{residuals},$$

where $SCP_{predicted}$ represents the variance SCP d.v. shares with beta power, and $SCP_{residuals}$ represents the portion of SCP d.v. that is independent from beta power.

$$beta_{predicted} = i + k*SCP_{total} + beta_{residuals},$$

where $beta_{predicted}$ represents the variance beta power shares with SCP d.v., and $beta_{residuals}$ represents the portion of beta power that is independent from SCP d.v.

We then compared how well beta power (magenta circle outline in **Fig 4A**) and beta power residuals (portion of magenta circle with "shared" part subtracted in **Fig 4A**; considered the SCP d.v.-independent portion of beta power) predicted perceptual outcome. Similarly, we compared how well SCP d.v. (purple circle outline in **Fig 4A**) and SCP d.v. residuals (portion of purple circle outline with "shared" part subtracted in **Fig 4A**) predicted perceptual outcome. To do so, we used the area under the receiver operating characteristic (ROC) curve, or the AUROC (see **Fig 4A**, lower panel). The ROC curve is a graph showing the performance of a classification model at all classification thresholds, and hence it quantifies how well a predictor can predict an outcome. The AUROC is such that a random classifier would have a value of 0.5, and a perfect classifier would have a value of 1.

If beta power and SCP d.v. share their influence on perceptual outcome, then the AUROC score for beta power residuals should be significantly lower than for beta power, and similarly, the AUROC score for SCP d.v. residuals should be significantly lower than for SCP d.v. The ROCs were constructed by using the class probabilities predicted by our models and shifting the discrimination threshold. We compared AUROCs for each measure and its residuals using a Wilcoxon signed-rank test, in each of the 2 beta clusters. We also conducted a Bayesian paired samples Wilcoxon *t* test to compare AUROCs, using JASP [89]. Using a one-sided Wilcoxon signed-rank test, we also verified that each variable's AUROC score was significantly higher than a random classifier's AUROC value of 0.5. We also conducted a Bayesian one-sample *t* tests, employing Cauchy prior widths of 0.707.

Similarly, we conducted an AUROC analysis comparing alpha power with alpha power residuals when beta power had been regressed out, and beta power with beta power residuals when alpha power had been regressed out. This was done for all alpha-beta cluster pairings of the low-level and high-level data sets.

**Pupil size analysis.** To assess to what extent each measure of spontaneous brain activity may be related to pupil-linked arousal, we correlated each measure with pupil size, using the data from the high-level visual task. We correlated SCP d.v. averaged across the entire pre-stimulus interval (−1.7 to −0.1 s) with pupil size averaged across the same interval. For alpha and beta power and their respective correlations with SCP d.v., we used estimates of all measures averaged across each cluster from the previous analysis, and hence across the pre-stimulus interval corresponding to each cluster. We obtained Spearman's rho and then assessed group-level statistics by testing the z-transformed rho values against zero with a Wilcoxon

signed-rank test. We also performed one-sample Bayesian $t$ tests using JASP [89], employing a Cauchy prior width of 0.707.

**Mediation analyses.** To understand the interrelation between each measure of pre-stimulus spontaneous activity, perceptual outcome, and pupil size, we utilized 3 mediation analyses.

In the first mediation analysis, we tested simultaneously whether pre-stimulus alpha power (variable A) may affect perceptual outcome (variable Y) via a modulation of pupil size (variable B), and whether pupil size (variable B) may affect perceptual outcome (variable Y) via a modulation of pre-stimulus alpha power (variable A). In other words, we tested both whether the influence of alpha power on perceptual outcome is mediated via pupil size, and/or whether the influence of pupil size on perceptual outcome is mediated via alpha power. For this analysis, and each mediation analysis described thereafter, we ran 5 regression models, as detailed below. In all equations, $i$ indicates the intercept term.

$$B = i + a*A \tag{1}$$

$$A = i + b*B, \tag{2}$$

where **a** and **b** are the linear regression coefficients reflecting the direct effect between A and B.

$$P(Y = 1|B) = \frac{1}{1 + e^{-(i+c*B)}}, \tag{3}$$

where **c** is the logistic regression coefficient reflecting the direct effect between B and Y, and where P(Y = 1|B) is the predicted probability that Y = 1 given B.

$$P(Y = 1|A) = \frac{1}{1 + e^{-(i+d*A)}}, \tag{4}$$

where **d** is the logistic regression coefficient reflecting the direct effect between A and Y, and where P(Y = 1|A) is the predicted probability that Y = 1 given A.

$$P(Y = 1|A, B) = \frac{1}{1 + e^{-(i+e*B+f*A)}}, \tag{5}$$

where **e** is the logistic regression coefficient reflecting the indirect effect between B and Y, given A, and **f** is the logistic regression coefficient reflecting the indirect effect between A and Y, given B, and P(Y = 1|A,B) is the predicted probability that Y = 1 given A and B.

To demonstrate that variable A's effect on variable Y is mediated by its effect on variable B, the following conditions must be true: **a**, **e**, and **d** must be significantly different from zero across subjects. That is, variable A should be significantly related to the outcome variable Y, resulting in a significant coefficient **d** in Eq 4. Secondly, variable A should be significantly related to the hypothesized mediating variable B, producing a significant coefficient **a** in Eq 1. Thirdly, the mediating variable B must be significantly related to the outcome variable Y, controlling for variable X, thus resulting in a significant coefficient **e** in Eq 5. Finally, to find evidence of mediation, then the relationship between variable A and outcome variable Y should be weaker when the mediating variable B is included in the model; that is, the influence of variable A on outcome variable Y should decrease when variable B is controlled for. Thus, the coefficient **f** should be smaller than coefficient **d**, i.e., **d – f > 0**. Conversely, to demonstrate that variable B's effect on variable Y is mediated by its effect on variable A, the following conditions must be true (following the same logic as above): **b**, **f**, and **c** must be significantly different from zero, and coefficient **e** should be smaller than **c**, i.e., **c – e > 0.**

This approach to the mediation analysis was followed exactly when testing for mediation between beta power, pupil size, and perceptual outcome, and between SCP d.v., pupil size, and perceptual outcome. In these cases, variable A was replaced by beta power, and by SCP d.v., respectively. All described analyses follow a causal step approach, whereby we analyzed the coefficients of 5 linear and logistic regression models [90–94].

Note that all continuous variables, i.e., pre-stimulus alpha power, pre-stimulus beta power, pupil size, and SCP d.v. were z-score normalized before input into linear and logistic regression models. As a result, all regression coefficients are standardized. A consequence of this is that **a** and **b** coefficients will be equal within each mediation analysis.

To assess significance of pre-requisite effects (i.e., **a, e, d** $> \mathbf{0}$ for the effect of variable A on outcome variable Y being mediated by variable B or **b, f, d** $> \mathbf{0}$ for the effect of variable B on outcome variable Y being mediated by variable A), we used one-sample $t$ tests against zero. To assess the significance of the mediation effect (i.e., **d – f** $> \mathbf{0}$ for evidence of mediation via variable B or **c – e** $> \mathbf{0}$ for evidence of mediation via variable A), we obtained the test statistic by computing the ratio of the mediation effect (**d – f** or **c – e**) to its standard error. We used the survival function of the Student's t-distribution to obtain the one-tailed $p$-value.

As a measure of effect size of the various models used for the mediation analysis, we extracted $R^2$ scores for the linear regression models, and AUROC scores for the logistic regression models. These are reported in **Fig 6D**.

See **Fig 6** for model descriptions and results.

## Supporting information

**S1 Text. Supporting information.**
(DOCX)

## Acknowledgments

We thank Martijn Wokke and Giulia Gennari for useful discussions and feedback on the manuscript.

## Author Contributions

**Conceptualization:** Biyu J. He.

**Formal analysis:** Lua Koenig.

**Funding acquisition:** Biyu J. He.

**Investigation:** Lua Koenig.

**Methodology:** Lua Koenig.

**Project administration:** Biyu J. He.

**Resources:** Biyu J. He.

**Supervision:** Biyu J. He.

**Validation:** Lua Koenig, Biyu J. He.

**Visualization:** Lua Koenig.

**Writing – original draft:** Lua Koenig, Biyu J. He.

**Writing – review & editing:** Lua Koenig, Biyu J. He.

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
