## [Editor Report · Decision Letter 0]

24 May 2024

Dear Biyu, 

Thank you for submitting your manuscript entitled "Spontaneous slow cortical potentials and brain oscillations independently influence conscious visual perception" for consideration as a Research Article by PLOS Biology.

Your manuscript has now been evaluated by the PLOS Biology editorial staff and I am writing to let you know that we would like to send your submission out for external peer review.

Once your full submission is complete, your paper will undergo a series of checks in preparation for peer review. After your manuscript has passed the checks it will be sent out for review. To provide the metadata for your submission, please Login to Editorial Manager (https://www.editorialmanager.com/pbiology) within two working days, i.e. by May 26 2024 11:59PM.

Kind regards,

Christian

Christian Schnell, PhD

Senior Editor

PLOS Biology

cschnell@plos.org

---

## [Decision Letter · Decision Letter 1]

30 Jul 2024

Dear Biyu,

Thank you for your patience while your manuscript "Spontaneous slow cortical potentials and brain oscillations independently influence conscious visual perception" was peer-reviewed at PLOS Biology. Apologies for the long delay in sending our decision, I struggled a bit with finding reviewers and an academic editor to handle your submission. However, your manuscript has now been evaluated by the PLOS Biology editors, an Academic Editor with relevant expertise, and by several independent reviewers. 

In light of the reviews, which you will find at the end of this email, we would like to invite you to revise the work to thoroughly address the reviewers' reports.

You will see below that the reviewers think that the study is very well executed and provides important insights. While the reports from Reviewer 1 and 3 are relatively short, they overlap with that of Reviewer 2, who submitted a very detailed report, outlining a couple of areas where you need to improve the analyses and interpretations, and provide some clarifications.

Given the extent of revision needed, we cannot make a decision about publication until we have seen the revised manuscript and your response to the reviewers' comments. Your revised manuscript is likely to be sent for further evaluation by all or a subset of the reviewers.

**IMPORTANT - SUBMITTING YOUR REVISION**

*Re-submission Checklist*

*Published Peer Review*

*PLOS Data Policy*

*Blot and Gel Data Policy*

Sincerely,

Christian

Christian Schnell, PhD

Senior Editor

PLOS Biology

cschnell@plos.org

REVIEWS:

Reviewer #1 (Afra Wohlschläger): The manuscript presents an interesting reanalysis of two MEG-datasets on upcoming visual perception on lower-level and higher-level visual stimuli, respectively. The influence of SCPs on perception is directly compared to the influence exerted by the oscillatory activities within the alpha- and beta-bands. The study finds significant but independent contributions to confidence in visual perception. For one study the relationship towards the noradrenergic system is additionally studied via the pupil diameter. This adds important insights on the mechanistic level.

The manuscript is well written and describing interesting and well-motivated analyses. 

I have few concerns:

1) Why are data on all stimuli used, i.e. why weren't catch trials omitted? Shouldn't this alter the interpretation in that prediction of confidence is assessed rather than perceptual success? The title should be:

'Spontaneous slow cortical potentials and brain oscillations independently influence confidence in visual perception'

Or alternatively a discussion of this aspect should be extended.

2) The analysis strategies differ between the two datasets with regards to the SCPs. I have related questions and remarks.

Methods section: Decoding analysis and Fig 2: How does the time axis in Fig. 2 A and D come about? The methods for deriving the d.v. are considerably different between the two data sets. From how I understand the analyses, for the low-level dataset there should be 1 d.v. value prior to the stimuli and none after, for the high-level dataset there should be four values prior to the stimuli and none after.

It should be made clear that Fig 1E only depicts the strategy for the lower-level dataset. Otherwise, the figures overall mimic identical strategies which is not the case.

Minor:

p.17 clustering analysis: time windows were treated independently so strictly a Bonferroni correction of 4 would need to be applied?

Please add to the citations:

On impact of LC activity on alpha rhythmic firing:

Dahl et al TICS (2022) Noradrenergic modulation of rhythmic neural activity shapes selective attention. DOI:https://doi.org/10.1016/j.tics.2021.10.009

On SCP influence on upcoming visual perception:

Glim et al Conscious Cogn (2020) The temporal evolution of pre-stimulus slow cortical potentials is associated with an upcoming stimulus' access to visual consciousness. DOI: 10.1016/j.concog.2020.102993 

Reviewer #2: The authors investigate the extent to which prestimulus oscillatory and non-oscillatory activity (slow cortical potential, SCP) independently or jointly predict reports of visual perceptual awareness. The current project emphasizes the particular relation between aperiodic fluctuation and lower-frequency (alpha/beta-band) oscillatory power in the prestimulus signal. This question is rooted in a large body of work that has examined the relation between prestimulus oscillatory activity and subsequent perception, where the focus has typically been on either oscillatory or SCP activity in isolation (with little inquiry regarding how these dynamics intersect). The current project extends this investigation through examination of the association between the two measures (applied in two independent samples performing distinct visual tasks), as well as their relationship with pupil-linked arousal (as measured through pupil size). The association between aperiodic fluctuation and neural oscillation is likely of current interest to a broad cross-section of researchers in the field of cognitive neuroscience, and in principle, I have no objections to the question or main features of the analytic approach. I do have some concerns related to the implementation (highlighted below), but I believe these are answerable.

The following are separated into major/minor comments listed as far as possible in order of appearance in the manuscript. Those concerns that I think most essential to address are noted parenthetically with "In the reviewer's opinion, essential to address."

Major concerns/questions:

Introduction:

Overall, the introduction is clear and well-written. However, there are a few areas that could benefit from more cautious writing. One general point is below, and a few specific notes are highlighted in the 'minor comments' section. 

Introduction, Paragraph 2 and throughout the document: 

The language choices of the authors regarding the relation between ongoing changes in EEG signal and perception are very causal. Although there certainly appear to be relations between oscillatory/low-frequency signal changes and perceptual variability, claims regarding degree of causality ('influences') are still active subjects of debate in the literature (as well as claims regarding which perceptual processes are impacted). The authors might consider adopting more cautious language surrounding the nature of the relation between prestimulus activity and perception (perhaps a choice as simple as shifting from 'influence' to 'may influence').

Method & Results: 

I apologize if my comments on method here seem lengthy. In part, this is due to the fact that the authors' main analytic goals depend on obtaining accurate representations of prestimulus SCP and oscillatory activity, so it seems important to consider any analytic choices/potential confounds that could impact measurement. 

Eye movement/blink preprocessing: 

The authors adopt the practice of removing eye movement-related artifacts with ICA, and rejecting trials in which blinks occurred during presentation. Both of these are standard and good-practice approaches to preprocessing in this context. However, I think the analyses here could be improved by including a more detailed assessment of blink-related activity. 

I have concerns on two broad counts. First, residual blink artifacts in the MEG signal can remain after ICA and would likely be within the frequency range targeted by SCP analyses. For this reason, it would be useful to understand if & to what extent the presence of blinks in the pre-stimulus window impacts the size of the low-frequency effects (meaning, when trials with detectable blinks in the prestimulus analysis window are rejected, do differences related to subsequent behavioral response remain?). 

Second, from both a cognitive and neural standpoint, there are ways in which blinks can impact subsequent awareness and neural activity beyond myogenic contamination of the MEG signal (see, for example, Kern et al., 2021 for an assessment of blink-related changes in intracranial occipital activity). It would therefore also be valuable to know whether frequency/timing of blinks in the prestimulus interval differs for 'seen' vs. 'unseen' targets (i.e. whether blink-related activity might reasonably explain either SCP or oscillatory effects, and possibly alter the apparent relation between them). This could be done for the second experiment most easily (given that eye-tracking data are not available for the first). This would also provide more context for the mediation results.

Wavelet decomposition: 

The justification for choosing a wavelet approach to signal decomposition here is a bit unclear. Given that wavelets necessarily involve a degree of temporal smearing (which the authors do make an effort to avoid), an alternate approach would be to compute the Fourier transform over each prestimulus window. This would avoid issues of temporal smearing, and also avoid the issue that the shape of the power spectrum is not the same when averaging over wavelet-transformed output vs. taking the simple FFT of data in the same window--which can impact the results of the 1/f decomposition. It seems preferable here on all counts (avoiding temporal smearing, desiring to perform a 1/f decomposition of the data) to adopt an FFT-based approach. If the authors are concerned about the sampling resolution of the first dataset, zero-padding to the same resolution as the second dataset would allow approximate comparison of frequency windows between datasets. 

Decomposition using Fooof & selection of trials (p.17; In the reviewer's opinion, essential to address): 

Theories regarding prestimulus activity fluctuations and the degree to which they do/do not predict subsequent reports of visual awareness acknowledge that oscillatory power can shift dramatically from trial to trial. These shifts are part of what is hypothesized to lead to changes in subsequent awareness. Accordingly, it doesn't seem justified to exclude a trial/timepoint simply because Fooof does not discover a peak for the spectrum in that window (this also seems problematic from a statistical perspective when trying to understand the relations between signals). Rather, if the authors are inclined to use 1/f decomposition, it seems like a safer approach would be to obtain more stable estimates of center frequency and bandwidth by deriving them from the overall signal for each electrode/subject/task. Then, apply these to the single-trial, 1/f-corrected spectra, ultimately obtaining even numbers of measurements for beta and alpha power for each timepoint/trial (i.e., avoiding variation in the number of trials/electrodes contributing to alpha-band, beta-band and SCP measurement). 

Relatedly, would the authors consider including a breakdown of how many trials remain for analysis for each subject/condition after all exclusions are performed? In addition, if the authors are disinclined to follow the suggestion above, it would be helpful to include information regarding the lack of inclusion of non-oscillatory trials in the main text of the document, since this is important to evaluation of the main analyses. 

Correction for 1/f activity in log vs. linear space (In the reviewer's opinion, essential to address): 

It is not entirely clear from the method section whether the authors perform 1/f correction in log space (the Fooof default) or linear space (after back-transforming aperiodic estimates into linear space). Each of these approaches implies different assumptions regarding how the sources of oscillatory and aperiodic data are combined at the scalp level, and if these assumptions are violated, the relation between aperiodic and oscillatory activity can be warped as a result. Since the primary aim of this paper is to relate changes in a low-frequency aperiodic pattern to changes in oscillatory power, it seems critically important to consider and explicitly justify these assumptions. To elaborate (perhaps too much):

Most individuals attempting separation of broadband and oscillatory activity assume, implicitly or explicitly, that the generative process is additive - that is, that broadband and oscillatory time series are additive at the level of the scalp (this is consistent with available evidence; e.g. Gyurkovics et al., 2021, among others). Parsing a power spectrum into oscillatory and non-oscillatory components then relies on the linearity property of the fourier transform, i.e. the spectrum of the sum of two signals is equal to the sum of their respective spectra (spectrum(oscillation + broadband) = spectrum(oscillation) + spectrum(broadband)). This implies that one can simply subtract the broadband spectrum from the aggregate spectrum to obtain a reasonably accurate impression of the spectrum of an oscillation.

However, many approaches to spectral separation (including Fooof) encourage separation of spectra in log or semi-log space, adopting a subtractive approach to decomposition. The logic is often presented as some form of: log(oscillation_spectrum + broadband_spectrum) - log(fitted_broadband_spectrum) = log(oscillation). But the additivity of spectra is not preserved over log transformation (i.e., log(a+b) != log(a) + log(b), and correspondingly, log(a+b) - log(b) != log(a)). If the generative process actually is additive, obtaining an accurate understanding of 'oscillation_spectrum' and 'broadband_spectrum' cannot be achieved by applying a subtractive correction in log space. Rather, subtraction in log-space is only justified if there is reason to believe that the generative process is multiplicative (i.e. log(oscillation*broadband) - log(broadband) = log(oscillation)). 

This is not just a theoretical issue. In our own data, we've seen adjustment in log-space result in dramatically different oscillatory effects (& relations between oscillatory power estimates and aperiodic fluctuation) than adjustment in linear space. We suspect these issues have contributed to a number of reports currently in the literature.

Here, it appears that the authors have (understandably, given the literature) adopted a log-space subtraction approach. If the authors believe multiplicative generation is the source of their spectra, this should be explicitly stated & justified. Otherwise, the correction approach should be modified to correct in linear space.

Merging 'real'/'catch' trials (Results, p4): 

The choice to merge both real/catch trials (and both intact/scrambled image trials) is a bit unusual. It seems like it would be more appropriate to separately examine target-present trials (first dataset) and intact images (second dataset). In both cases, the choice to merge trials from both conditions means that what is actually measured is not really whether prestimulus power predicts perceptual experience, but rather the likelihood of making a 'seen' response regardless of stimulus. In the case of the first dataset, there are so few catch trials that the decision to include them or not may not ultimately impact the conclusions drawn here. However, in the second case, 60 scrambled 'catch' trials is not an insignificant number. Given that there is evidence that pre-stimulus activity (particularly in the alpha range) may impact some aspects of decision-making but not others (e.g. Iemi et al., 2017; Samaha et al., 2020) it seems very inadvisable to lump 'hits' and 'false alarms' into the same data pool without interrogation. If the only justification for pooling this data is 'statistical power', then there should be some demonstration that the pattern of results does not qualitatively differ between intact/scrambled images. 

In addition, the claim "As such, we focus on whether different pre-stimulus neural activities have shared influences on perceptual outcome, instead of dissociating pre-stimulus activity's influence on sensitivity and criterion related to conscious perception as previous studies have done." However, the choice to pool hits/false alarms into a single category essentially means that the authors are examining effects likely related to criterion. 

SCP vs. oscillatory analyses:

For analyses of SCP and pre-stimulus power, I am not sure I follow why a classification approach over all electrodes is adopted for the SCP data, and a simple power comparison is conducted for alpha and beta-power analyses over a smaller range of timepoints. To maximize capacity to relate SCP dynamics to higher-frequency oscillatory power dynamics, I would think it preferable to maximize similarity of analytic approaches here. Perhaps I'm missing some feature of the analysis that necessitates this approach. Could the authors clarify why they have adopted two distinct approaches here?

Correlation analyses: 

This is optional, but to better capture the hierarchical structure of the data (association between trial-wise alpha power and SCP, for example, nested within subject), it might be useful here to generate mixed-effects models with subject as a random effect (instead of correlating within subject and then performing a group-wise analysis over the correlation coefficients).

Interpretation of Correlations & Bayesian tests (p6; In the reviewer's opinion, essential to address): 

The interpretation of Bayes Factors here is a bit unconventional (although this seems to be an issue associated with the correlation analyses and not the rest of the analyses in the document). A Bayes Factor of 1 corresponds to ambiguous evidence for/against either of two hypotheses. Although criteria for evaluating magnitude of Bayes Factors are somewhat arbitrary, the convention in much of the literature is to treat Bayes Factors of between ⅓ and 3 as providing weak/ambiguous evidence. So, a Bayes Factor of .828 (very nearly 1) corresponds to at best ambiguous evidence very weakly favoring the null hypothesis of no relation. It should not be considered moderate evidence, and certainly doesn't suggest a lack of correlation. Similarly, a Bayes Factor of .331 would most often be considered weak evidence in favor of the null hypothesis, and a BF of 1.24 would probably best be considered essentially ambiguous. 

Alpha power & SCP, paragraph 4 - "Together these results suggest" - The authors might want to revisit the conclusions in this paragraph, which do not reflect the evidence just provided. Phrases like "did not co-vary" and "must be independent" are too strong. A non-significant p-value does not imply a total lack of shared variance, and the Bayes Factors provide at best weak/ambiguous evidence in either direction. It would be more accurate to say something like "although each spontaneous signal was individually linked to subsequent behavior, there was not evidence of a strong association between the signals." ` 

Beta power and SCP, paragraph 2 - "This result suggests" - As above, it is incorrect to claim that a lack of significant correlation implies independence, or that these results imply that these two signals "do not have any covariation."

Beta power and SCP, paragraph 3: The authors are correct that Bayes Factors of 1.835 and 2.777 provide at best weak evidence in favor of the alternative hypothesis (for some additional elaboration on Bayes Factor interpretation, see Kass & Raftery, 1995). As such, I don't think it's entirely accurate to use these Bayes Factors to support the statement that "the correlations between beta power and SCP d.v. were different from 0" (this statement is maybe better placed in reference to the initial p-values).

Discussion: 

As with the introduction, the discussion is generally clear and well-written. I have just one comment on language use that I think could improve the document. In paragraph 1 (and throughout), the authors repeatedly make claims about independence that seem too strong/incautious relative to the data presented. The data might support claims about possible or partial independence of mechanism, but to imply that oscillatory and aperiodic activity patterns reflect totally independent mechanisms linked to perception is a bit beyond what the data support. The authors might consider re-evaluating the discussion section with an eye toward more careful language.

Minor comments:

Abstract: This is relatively minor, but the statement 'We found that oscillatory power and large-scale SCP activity influence conscious perception through independent mechanisms that do not have shared variance' could be written more cautiously given the design. Use of 'influences' implies a known causal brain-behavior relation that is not clearly warranted given the literature and the correlational structure of the data. Second, and perhaps more importantly, 'do not have shared variance' is probably too strong given the results, which don't imply a totally orthogonal relationship between the two signal features. 

Introduction, paragraph1, sentence 3: The authors might consider revisiting the two examples provided, as they are not clear examples of the argument 'This variability shows that conscious perception is not determined solely by the incoming sensory information.' Variability in perception is probably better highlighted as variability in experience of the same physical stimulus under different circumstances. The current examples are of two very different stimuli, and it could easily be argued that the stop sign is more salient to a driver than an existing crack in the windshield (which might not be salient at all to a driver used to its presence). This is not a strictly necessary change, but if the language distracted this reviewer, it seems likely to distract other readers too.

Introduction, paragraph 3: 'perceived or missed' - the authors might consider revisiting the choice of 'perceived' here, since although alpha power dynamics seem to predict subjective reports of awareness, the evidence that they actually predict the subsequent percept is at best debatable (e.g. Iemi et al., 2017). 

Method, EEG preprocessing (both datasets): This is optional, but it would be helpful for the reader if the data preprocessing (or results section in the main document) included a statement regarding the extent to which preprocessing/analysis steps were identical/differed between these analyses and prior published studies with these datasets.

Classification analyses: Could the authors clarify the rationale for applying two distinct approaches to classification for the low- and high-level datasets? It seems preferable when looking across two datasets in the same paper to apply the same approach in both cases as far as possible, although admittedly not necessary since datasets are treated independently throughout. 

1/f decomposition: The figure illustrating aperiodic decomposition appears remarkably similar to one used by the authors of Fooof on their website (https://fooof-tools.github.io/fooof/auto_examples/plots/plot_fooof_models.html#sphx-glr-auto-examples-plots-plot-fooof-models-py). If this figure was inspired by or adapted from their work, it seems warranted to credit them in the caption.

Results, Paragraph 4 of 'Pre-stimulus alpha power predicts perceptual outcome': The authors might consider discussing how these results (in the higher level dataset) could relate to prior observations in the literature of differing pre-stimulus dynamics related to responses in detection vs. discrimination contexts (e.g. Iemi et al., 2017).

Figure 2: If possible, it would be helpful to have some sense of the topographic distribution of prestimulus SCP activity in both tasks, so as to better relate this to the observed clusters of alpha/beta power.

Beta power and SCP, final paragraph: If I understand the AUROC analyses correctly, the statement "This demonstrates that pre-stimulus beta power and SCP d.v. explain non-overlapping variance in the perceptual outcome" may not be strictly correct. These analyses demonstrate that non-shared components of variation in beta power and SCP can each predict subsequent perceptual outcome, but I don't think this necessarily implies that the variance in perceptual outcome that they both predict is itself non-overlapping.

Discussion, general: The authors might consider discussion of how their results relate to the observations of Gyurkovics et al. 2021 (The impact of 1/f activity and baseline correction on the results and interpretation of time-frequency analyses of EEG/MEG data: A cautionary tale) that alpha-band power is not positively correlated with 1/f 'background' (See also single-trial correlational results reported by Cunningham et al., 2023). 1/f slope and offset in the EEG can be influenced by amplitude of low-frequency activity, so it seems worth acknowledging or considering how these existing results might or might not reflect similar observations to those described here.

Discussion P1: 'Importantly, our main findings were reproduced in two separate datasets involving different visual stimuli, showing the reproducibility and generalizability of the present findings.' I think this statement might be more misleading than the authors intend, since it seems to refer to their prior statement in this paragraph regarding pupil-linked arousal (the statement seems to suggest both pupil-linked and 'independence of signal' findings were fully generalized across two datasets). The authors might consider tweaking the wording to emphasize that 'main findings' refers to the relation between oscillatory power and SCP. 

Reviewer #3: Thank you for inviting me to review this manuscript by Koenig and He, in which the authors interrogate the relationship between low and high frequency MEG signals and two distinct threshold detection paradigms. Using a decoding-based approach, the authors demonstrate independent evidence for low and high frequency signals prior to stimulus presentation, which they then relate to the ascending arousal system through a statistical relationship with pupil diameter.

This study was well-motivated, clearly-presented and discussed within the context of a broad literature. I have only minor comments, which I hope are helpful.

* P4 - I'm not familiar with the term "temporal spearing". Please adjust or define.

* Methods - as written, I worry that the reader may get the impression that statistical decodability is synonymous with conscious perception, whereas it's possible that signals were present in the pre-stimulus window that were related to the primary processing, but not the conscious perception of the stimuli. In my opinion, the manuscript would benefit from this point being made clearly for the reader.

* P10 - is it possible to create a model that combines alpha, beta and SCP with pupil size to determine their unique/combined perceptual influences?

---

## [Decision Letter · Decision Letter 2]

15 Nov 2024

Dear Biyu,

Thank you for your patience while we considered your revised manuscript "Spontaneous slow cortical potentials and brain oscillations independently influence conscious visual perception" for publication as a Research Article at PLOS Biology. This revised version of your manuscript has been evaluated by the PLOS Biology editors, the Academic Editor and one of the original reviewers.

Based on the reviews and on our Academic Editor's assessment of your revision, we are likely to accept this manuscript for publication, provided you satisfactorily address the following data and other policy-related requests:

* Please add the links to the funding agencies in the Financial Disclosure statement in the manuscript details.

* Please note that per journal policy, the model system/species studied should be clearly stated in the abstract of your manuscript. 

* DATA POLICY:

Regardless of the method selected, please ensure that you provide the individual numerical values that underlie the summary data displayed in the following figure panels as they are essential for readers to assess your analysis and to reproduce it: 1BD, 2CF, 3BD, 4BC, 5AB, 6ABC and 7ABC.

* CODE POLICY

We expect to receive your revised manuscript within two weeks. 

*Published Peer Review History*

*Press*

Sincerely,

Christian

Christian Schnell, PhD

Senior Editor

cschnell@plos.org

PLOS Biology

Reviewer remarks:

Reviewer #2: I want to thank the authors sincerely for their thorough and considered response response to what was, admittedly, a lengthy review. I have reviewed the responses and the revised document, and I am satisfied that my concerns have been addressed. Thank you for the opportunity to review this quite interesting report!

---

## [Editor Report · Decision Letter 3]

3 Dec 2024

Dear Dr He,

Thank you for the submission of your revised Research Article "Spontaneous slow cortical potentials and brain oscillations independently influence conscious visual perception" for publication in PLOS Biology. On behalf of my colleagues and the Academic Editor, Christopher Pack, I am pleased to say that we can in principle accept your manuscript for publication, provided you address any remaining formatting and reporting issues. These will be detailed in an email you should receive within 2-3 business days from our colleagues in the journal operations team; no action is required from you until then. Please note that we will not be able to formally accept your manuscript and schedule it for publication until you have completed any requested changes.

When you attend to those requests, please also modify the reference to the source data in the figure legends by including the link to the repository in each figure legend. This way, readers to not have to look at multiple places to go to the source data.

PRESS

Sincerely, 

Christian

Christian Schnell, PhD

Senior Editor

PLOS Biology

cschnell@plos.org